# Implicit Shape Model Trees: Recognition of 3-D Indoor Scenes and Prediction of Object Poses for Mobile Robots

**Pascal Meißner** [1,*,†] and **Rüdiger Dillmann** [2]

1   School of Engineering, University of Aberdeen, Aberdeen AB24 3UE, UK
2   Humanoids and Intelligence Systems Lab (HIS), Karlsruhe Institute of Technology (KIT),
    76131 Karlsruhe, Germany; dillmann@kit.edu
*   Correspondence: pascal.meissner@thws.de
†   Current address: Center for Artificial Intelligence and Robotics (CAIRO), Technical University of Applied
    Sciences Wuerzburg-Schweinfurt (THWS), 97082 Wuerzburg, Germany.

**Abstract:** This article describes an approach for mobile robots to identify scenes in configurations of objects spread across dense environments. This identification is enabled by intertwining the robotic object search and the scene recognition on already detected objects. We proposed "Implicit Shape Model (ISM) trees" as a scene model to solve these two tasks together. This article presents novel algorithms for ISM trees to recognize scenes and predict object poses. For us, scenes are sets of objects, some of which are interrelated by 3D spatial relations. Yet, many false positives may occur when using single ISMs to recognize scenes. We developed ISM trees, which is a hierarchical model of multiple interconnected ISMs, to remedy this. In this article, we contribute a recognition algorithm that allows the use of these trees for recognizing scenes. ISM trees should be generated from human demonstrations of object configurations. Since a suitable algorithm was unavailable, we created an algorithm for generating ISM trees. In previous work, we integrated the object search and scene recognition into an active vision approach that we called "Active Scene Recognition". An efficient algorithm was unavailable to make their integration using predicted object poses effective. Physical experiments in this article show that the new algorithm we have contributed overcomes this problem.

**Keywords:** part-based models; Hough transform; spatial relations; object arrangements; object search; mobile robotics





## 1. Introduction

To act autonomously in various situations, robots not only need the capabilities to perceive and act, but must also be provided with models of the possible states of the world. If we imagine such a robot as a household helper, it will have to master tasks such as setting, clearing, or rearranging tables. Let us imagine that such a robot looks at the table in Figure 1 and tries to determine which of these tasks is pending. More precisely, the robot must choose between four different actions, each of which contributes to the solution of one of the tasks. An autonomous robot may choose an action based on a comparison of its perceptions with its world model, i.e., its assessment of the state of the world. Which scenes are present is an elementary aspect of such a world state. Modeling scenes and comparing their models with perceptions is the topic of this article. In particular, we model scenes not by the absolute poses of the objects in them, but by the spatial relations between these objects. Such a model can be more easily reused across different environments because it models a scene regardless of where it occurs.

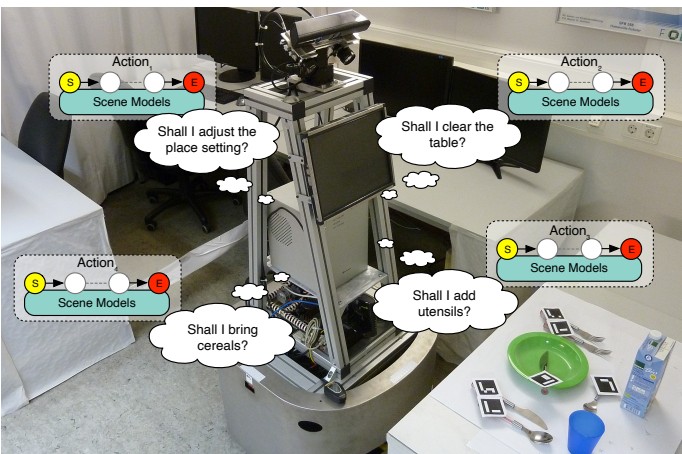

**Figure 1.** Motivating example for scene recognition: Our mobile robot MILD looking at an object configuration (a.k.a. an arrangement). It reasons which of its actions to apply.

## 1.1. Scene Recognition—Problem and Approach

This article provides a solution to the problem of classifying into scenes a configuration (a.k.a. an arrangement) of objects whose poses are given in six degrees of freedom (6-DoF). Recognizing scenes based on the objects present is an approach suggested for indoor environments by [1] and successfully investigated by [2]. The classifier we propose not only describes a single configuration of objects, but rather a multitude of configurations that these objects can take on while still representing the same scene. Hereinafter, this multitude of configurations will be referred to as a "scene category" rather than as a "scene", which is a specific object configuration. For Figure 1, the classification problem we address can be paraphrased as follows: "Is the present tableware an example of the modeled scene category?", "How well does each of these objects fit our scene category model?", "Which objects on the table belong to the scene category?", and "How many objects are missing from the scene category?" Our classifier is learned from object configurations demonstrated by a human in front of a robot and perceived by the robot using 6-DoF object pose estimation.

Many scene categories require that the spatial characteristics of relations, including uncertainties, be accurately described. For example, a table setting requires that some utensils be exactly parallel to each other, whereas their positions relative to the table are less critical. To meet such requirements, we proposed single implicit shape models (ISMs) as scene classifiers in [3]. Inspired by Hough voting, a classic but still popular approach (see [4,5]), our ISMs let each detected object in a configuration vote on which scenes it might belong to, thus using the spatial relations in which the object participates. The fit of these votes yields a confidence level for the presence of a scene category in an object configuration. Overall, this article is not about feature extraction, but about modeling relations and their variations in 6-DoF. ISMs for scene recognition should not be considered as an alternative, but rather as a complement to the immensely successful convolutional neural nets (CNNs).

## 1.2. Object Search—Problem and Approach

Figure 2 shows our experimental kitchen setup as an example of the many indoor environments where objects are spatially distributed and surrounded by clutter. A robot will have to acquire several points of view before it has observed all the objects in such a scene. This problem is addressed in a field called three-dimensional object search ([6]). The existing approaches often rely on an informed search. This method is based on the fact that detected objects specify areas where other objects should be searched. However, these areas are predicted by individual objects rather than by entire scenes. Predicting poses utilizing individual objects can lead to ambiguities, since, e.g., a knife in a table setting would expect the plate to be beneath itself when a meal is finished, whereas it would expect the plate to

be beside itself when the meal has not yet started. Instead, using estimates for scenes to predict poses resolves this problem.

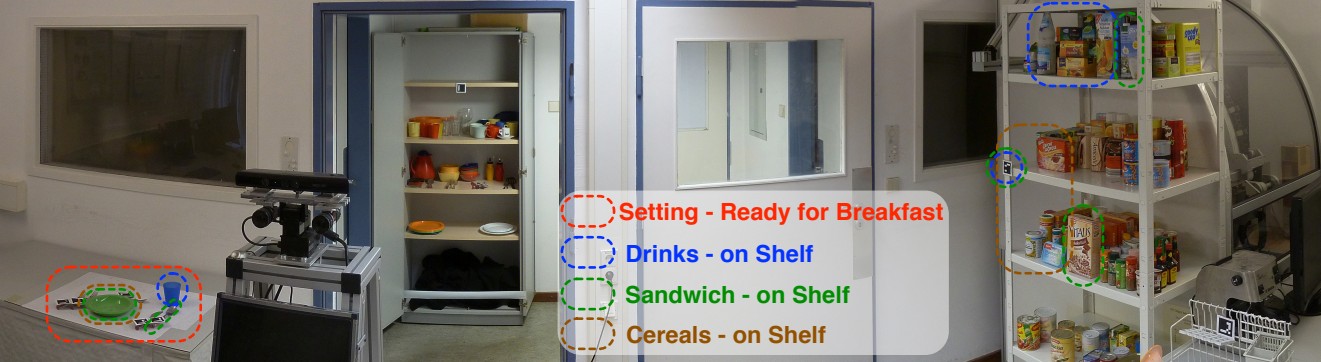

**Figure 2.** Experimental setup mimicking a kitchen. The objects are distributed over a table, a cupboard, and some shelves. Colored dashed boxes are used to discern the searched objects from the clutter and to assign objects to exemplary scene categories.

To this end, we presented 'Active Scene Recognition' (ASR) in [7,8], which is a procedure that integrates the scene recognition and object search. Roughly, the procedure is as follows: The robot first detects some objects and computes which scene categories these objects may belong to. Assuming that these scene estimates are valid, it then predicts where missing objects, which would also belong to these estimates, could be located. Based on these predictions, camera views are computed for the robot to check. In this article, ASR is detailed in Section 4. In Section 5.3, we evaluate ASR and this article's contributions to it on a physical robot. To distinguish between ASR and pure scene recognition, the latter is referred to as 'Passive Scene Recognition' (PSR). PSR is detailed in Section 3. The flow of our overall approach (see [9]), which consists of two phases—first the learning of scene classifiers and then the execution of active scene recognition—is shown in Figure 3.

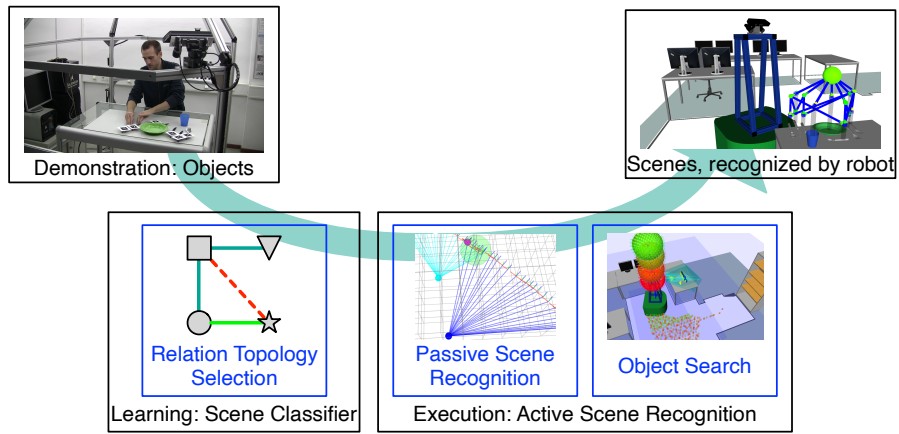

**Figure 3.** Overview of the research problems (in blue) addressed by our overall approach. At the top are the inputs and outputs of our approach. Below are the two phases of our approach: scene classifier learning and ASR execution.

### 1.3. Relation Topology Selection—Problem

We train our scene classifiers using sensory-perceived demonstrations (see [10]), which consist of a two- to three-digit number of recorded object configurations. This learning task involves the problem of selecting pairs of objects in a scene to be related by spatial relations. Which combination of relations is modeled determines the number of false positives returned by a classifier and the runtime of the scene recognition. Combinations of relations are hereinafter referred to as relation topologies. Whereas such topologies contain

only binary relations, they can represent many ternary or n-ary relations with multiple binary ones. In Section 3.8, we justify and outline how we selected the relation topologies in our previous work in [11].

### 1.4. Contributions and Differences from Previous Work

To not unnecessarily restrict this topology selection, a scene classifier should be able to represent a maximum of topologies. To this end, we first suggested 'Implicit Shape Model trees', which is a hierarchical scene model, in [3]. This model consists of multiple ISMs stacked upon each other, with the ISMs in it representing different portions of the same scene category. Such portions are brought together by additional ISMs in such a tree. The closer an ISM is to the root of a tree, the larger the portion it covers. However, while we outlined such a tree in [3], we did not define how the scene recognition in terms of data or control flow would work with ISM trees. In Section 3.7, we close this gap, which prevents greater use of ISM trees by contributing an algorithm for scene recognition with ISM trees. In [11], we defined how to select relation topologies. However, we did not describe how ISM trees are generated from such topologies. In Section 3.6, we contribute an algorithm for generating ISM trees, thus closing another essential gap.

As is visible in Figure 3, learned ISM trees are used to perform ASR. To make ASR possible, we had to link two research problems: scene recognition and object search. To this end, we proposed a technique for predicting the poses of searched objects in [7] and reused it in [8]. However, this technique suffered from a combinatorial explosion. We close this gap, which made ASR impractical for larger scenes, by contributing a prediction algorithm in Section 4.2 that efficiently predicts object poses. In summary, this work's contributions are as follows:

1. We created an algorithm for generating ISM trees.
2. We created an algorithm for scene recognition using ISM trees.
3. We created an algorithm for predicting object poses using ISM trees.

### 1.5. Equipment and Constraints

For the experiments with our robot MILD in Figure 1, we integrated these algorithms into ASR. The robot consists of a mobile base and a pivoting camera head. Searched objects are detected using third-party object pose estimators. From all the searched objects in our kitchen setup, only the utensils are localized using markers. ISM trees provide two parameters that set the degree to which object poses may deviate from the modeled relations without being excluded from the scene. These parameters were tested in the range of [mm] to [dm] for object positions and in the one- to two-digit [°] range for object orientations. Because ISM trees emphasize the modeling of relations, they focus on the objects in a scene. They complement works like that of [12] that placed emphasis on the global shape of a scene, including the walls or the floor. ASR can only search objects that are part of demonstrated scene categories. ASR also assumes that the environment is static during the object search, as opposed to approaches ([13]) that address dynamic scenes.

## 2. Related Work

### 2.1. Scene Recognition

The research generally defines scene understanding as an image labeling problem. There are two approaches to address it. One derives scenes from detected objects and relations between them, and the other derives scenes directly from image data without intermediary concepts such as objects, as [14] has investigated. Descriptions of scenes in the form of graphs (modeling existing objects and relation types), as derived by an object-based approach, are far more informative for further use, e.g., for mapping ([15,16]) or object search, than the global labels for images that are instead derived using the "direct" approach. Work in line with [17] or [18] (see [19] for an overview) that follows the object-based approach relies on neural nets for object detection, including feature extraction (e.g., through the work by [20,21]), wherein they combined it with neural nets to generate scene

graphs. This approach is made possible by datasets that include relations ([22,23]), which have been published in recent years alongside object detection datasets ([24,25]). These scene graph nets are very powerful, but they are designed to learn models of relations that focus on relation types or meanings rather than the spatial characteristics of relations. In contrast, in our work, we want to focus on accurately modeling the spatial properties of relations and their uncertainties. Nevertheless, our model should be able to cope with small amounts of data, since we want it to learn from demonstrations of people's personal preferences concerning object configurations. Indeed, users must provide personal data, wherein they tend to put in a limited effort.

Examples of preferences in object configurations can be breakfast tables, which only few people will want to have set in the same way. Nevertheless, people will expect household robots to consider their preferences when arranging objects. For example, ref. [26] addressed personal preferences by combining a relation model for learning preferences for arranging objects on a shelf with object detection. However, while their approach could even successfully handle conflicting preferences, it also missed subtle differences between spatial relations in terms of their extent. Classifiers explicitly designed to model relations and their uncertainties, such as the part-based models [27] from the 2000s, are a more expressive alternative. They also have low sample complexity, thus making them suitable for learning from demonstrations. By replacing their outdated feature extraction component with CNN-based object detectors or pose estimators (e.g., DOPE [28], PoseCNN [29], or CenterPose [30]), we obtain an object-based scene classifier that combines the power of CNNs with the expressiveness of part-based models in relation modeling. Thus, our approach combines pretrained object pose estimators with the relation modeling of part-based models.

Ignoring the outdated feature extraction of part-based models, we note that [31] already successfully used a part-based model, the constellation model [32], to represent scenes. Constellation models define spatial relations using a parametric representation over Cartesian coordinates (a normal distribution), just like the pictorial structures models [33] (another part-based model) do. Recently, ref. [34]'s approach of using probability distributions over polar coordinates to define relations has proven to be more effective for describing practically relevant relations. Whereas such distributions are more expressive than the model in [26], they are still too coarse for us. Moreover, they use a fixed number of parameters to represent relations. What would be most appropriate when learning from demonstrations of varying length is a relation model whose complexity grows with the number of training samples demonstrated, i.e., a nonparametric model [10]. Such flexible models are the implicit shape models (ISMs) of [35,36]. Therefore, we chose ISMs as the basis for our approach. One shortcoming that ISMs have in common with constellation and pictorial structures models is that they can only represent a single type of relation topology. However, the topology that yields the best tradeoff between the number of false positives and scene recognition runtime can vary from scene to scene. We extended the ISMs to our hierarchical ISM trees to account for this. We also want to mention scene grammars ([37]), which are similar to part-based models but motivated by formal languages. Again, they model relations probabilistically and only use star topologies. For these reasons, we chose ISMs over scene grammars.

### 2.2. Object Pose Prediction

The research addresses the search for objects in 3D either as an active vision ([38–43]) or as a manipulation problem ([44–49]). Active vision approaches are divided into direct ([50,51]) and indirect ([52]) searches depending on the type of knowledge about the potential object poses used. An indirect search uses spatial relations to predict from the known poses of objects those of searched objects. An indirect search can be classified according to the type ([53]) of relations used to predict the poses. Refs. [54–56], for example, used the relations corresponding to natural language concepts such as 'above' in robotics. Even though such symbolic relations provide high-quality generalization, they can only

provide coarse estimates of metric object poses—too coarse for many object search tasks. For example, ref. [57] successfully adapted and used Boltzmann machines to encode symbolic relations. Representing relations metrically with probability distributions showed promising results in [58]. However, their pose predictions were derived exclusively from the known locations of individual objects, thereby leading to ambiguities that the use of scenes can avoid.

### 3. Methods—Passive Scene Recognition

*3.1. Overview and ISMs as Described by Leibe et al. ([35,36])*

In Section 1, we introduced scene recognition as a black box that receives estimates for objects as input. From these, it derives as output the present instances of scene categories. The concrete process of how our approach to scene recognition works is shown in Figure 4. Firstly, external object pose estimators derive the types and poses of the present objects. Hence, a physical object configuration is transformed into a set of 'object estimates'. These estimates are passed on to each of our scene classifiers. Every scene classifier then returns estimates for the presence of instances (the confidence level of such an instance is visualized by a sphere above the scene. Its color changes from red to green when the confidence increases. Relations are shown as lines whose colors indicate to which ISM they belong) of one scene category, including where these instances are located in 3D space. Each scene classifier models a scene category and is learned from a recording of a human demonstration. Such a demonstration of object configurations is shown in **2** in Figure 5. A plate and a cup are pushed from left to right, thus yielding two parallel trajectories (trajectories are visualized as line strips between coordinate frames that stand for object poses). Please note that we used pretrained object pose estimators to record object poses during demonstrations. One of the demonstrated configurations is visible in **1** in Figure 5.

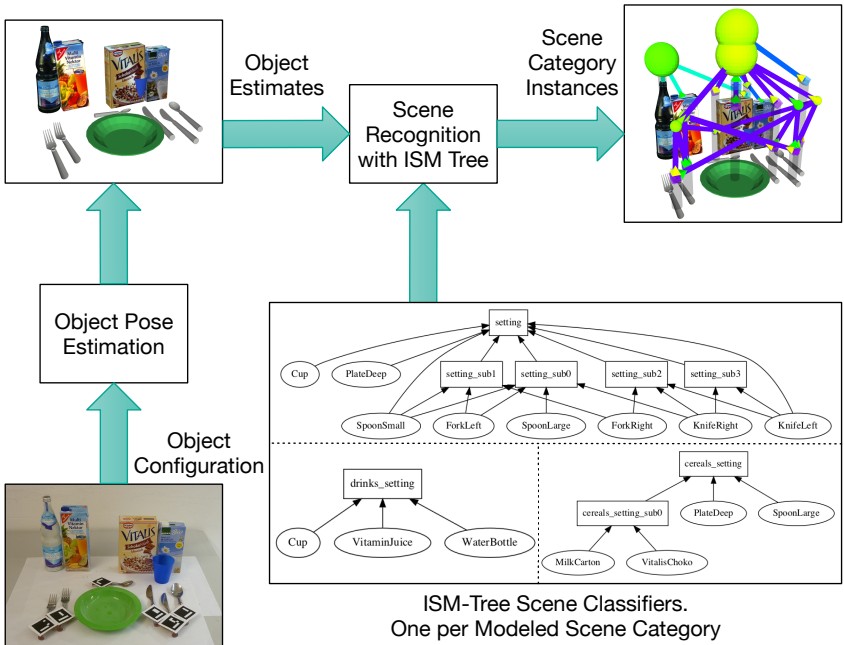

**Figure 4.** Overview of the inputs and outputs of passive scene recognition. Spheres represent increasing confidences of outputs by colors from red to green.

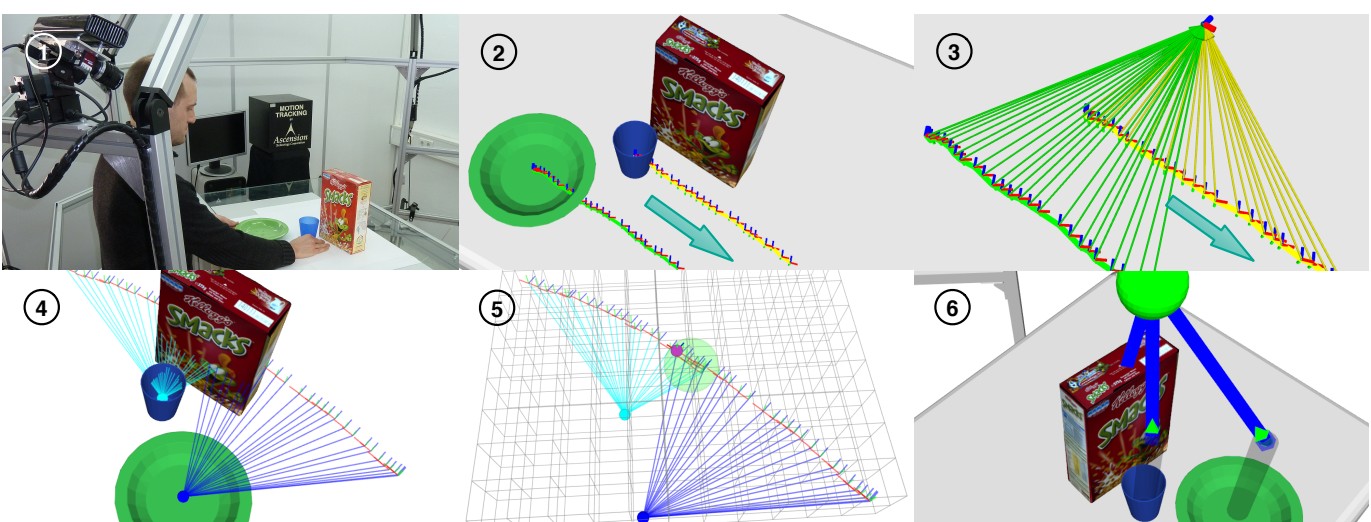

**Figure 5.** Image **1**: Snapshot of a demonstration. Image **2**: Demonstrated object trajectories as sequences of object estimates. The cup is always to the right of the plate, and both in front of the box. Image **3**: Relative poses, visualized as arrows pointing to a reference, form the relations in our scene classifier. Here, the classifier models the relations "cup-box" and "plate-box". Image **4**: Objects voting on poses of the reference using the relations from image **3**. Image **5**: Accumulator filled with votes. Image **6**: Here, the cup is to the left of the plate, which contradicts the demonstration. The learned ISM does not recognize this, as it does not model the "cup-plate" relation. It outputs a false positive.

In [3], we redefined implicit shape models (ISMs) so that they would represent scenes (which consist of objects) instead of objects (which consist of object parts). The original ISMs were used to recognize objects in 2D images and were similar to the generalized Hough transform ([59]). As [60] write, this transform is used to "locate arbitrary shapes with unknown position, size and orientation" in images. It does so by gathering evidence about the said properties of a shape from the pixel values in an image. Evidence gathering is implemented by learning a mapping from pixels to shape parameters for each shape. Using this mapping, pixels cast votes in an accumulator array for different parameter values. The parameter values of the present shapes can then be determined from the local maxima in this array.

### 3.2. Single ISMs as Scene Classifiers—Previous Work

The learning ([3]) of a similar mapping for our 'scene classifier ISMs' can be thought of as adding entries to a table, which is similar to k-nearest neighbors ([61]). In [3], we transformed the absolute poses in the trajectories of interrelated objects into relative poses, which we then stored in a table. Thus, the learning did not involve the optimization of the parameters of a model. Instead, the ISMs represented spatial relations as sequences of 6-DoF relative poses. Nevertheless, scene recognition could be done efficiently, as it mainly consisted of highly parallelizable matrix operations. We did not arbitrarily decide which pairs of absolute poses to convert into relative poses. Instead, among the objects in a scene category, we did set one as the reference object to which all relative poses pointed. In **2** in Figure 5, for example, the cornflakes box is the reference. Accordingly, all relative poses in **3** in Figure 5 (each relation consists of all visualized arrows of one color) point from the plate and the cup to the box.

Like the generalized Hough transform, scene recognition ([3]) with a single ISM started with a vote. Instead of letting pixels vote, the known objects cast votes starting from the place where they had been located. The voting was accomplished by combining the estimated pose of the respective object with all those relative poses in the table of the ISM that were assigned to an object. The visualization of a vote in **4** in Figure 5 shows at which poses the plate and the cup respectively expect the box. The votes cast were entered into the 3D accumulator array shown in **5** in Figure 5. Once the voting was completed, we

searched this array for the most comprehensive and consistent combinations of votes from the objects in a scene category. We identified the top-rated combinations as instances of the scene category that the ISM modeled. Note that it did not matter whether the reference object was present in that combination and that a missing reference did not cause the recognition to fail.

To avoid a combinatorial explosion during this search, we only compared votes that had fallen into the same bin of the accumulator by using a method similar to the mean shift search proposed by [36]. This procedure allowed for discarding votes from irrelevant objects, i.e., objects that did not belong to the modeled scene category. Since a single ISM only ever relates one reference object to all other objects in a scene, it can only represent a star-shaped topology of relations. This could lead to false positives in scene recognition as long as only relations between the other objects would be violated. For example, in **6** in Figure 5, we swapped the cup and the plate. Nevertheless, the ISM considered this configuration to be a valid instance of its scene category. Hence, star-shaped topologies and single ISMs are not sufficient to reliably recognize many scene categories.

### 3.3. Implicit Shape Model Trees—Outline

Instead, we create a scene classifier that supports all the connected relation topologies by first partitioning the given relation topology into star-shaped subtopologies, which are then assigned to separate ISMs. Based on this partitioning, we assemble the ISMs and connect them into a tree, thus creating a compound hierarchical model of the initial relation topology: The ISM tree. To avoid a combinatorial explosion when using an ISM tree for scene recognition, we take the precaution that only a restricted amount of data, the most comprehensive and consistent combinations of votes in each ISM, is shared between connected ISMs. Such an approach could have led to false negatives in scene recognition. However, such an effect was not observed during our experiments in Section 5. Before we detail this article's contributions to the ISM trees, we present the assumptions and notation used throughout the article in Section 3.4 and outline a technique from our previous work that partitions connected topologies into stars in Section 3.5. Section 3.6 introduces a novel algorithm for generating ISM trees from these stars. As yet another contribution, we present an algorithm for recognizing scenes with ISM trees in Section 3.7.

### 3.4. Preliminaries—Definitions for Scene Recognition

We define an object $o$ as an entity whose state $\mathbf{E}(o, t) = (c, d, \mathbf{T})$ at a point in time $t$ is estimated from sensor data. The state is described by a triple consisting of a label $c$ indicating the object class, a label $d$ used to distinguish between different objects of the same class, and a transformation matrix $\mathbf{T} \in \mathbb{R}^{4 \times 4}$ indicating the pose of the object. A scene category $\mathbf{S} = (\{o\}, \{\mathbf{R}\})$ consists of objects and the spatial relations $\{\mathbf{R}\}$ between the objects. The identity of a scene category is defined by a label $z$, and each spatial relation is represented as a set of relative 6-DoF poses $\{\mathbf{T}_{jk}\}$. In scene recognition, the fit between a configuration $\{\mathbf{E}(o, t)\}$ of objects (a set of states) and the model of a scene category is estimated. If this fit, whose degree is indicated by a confidence level $b(\mathbf{I_S}) \in [0, 1]$, is sufficiently good, we consider the objects as an instance $\mathbf{I_S}$ of the scene category and locate it at a pose $\mathbf{T}_F$. Models of scene categories are learned from trajectories demonstrated over $l$ time steps for each object included in the category. Each trajectory is a sequence $\mathbf{J}(o) = (\mathbf{E}(o, 1), \ldots, \mathbf{E}(o, l))$ of estimates of the time-variant state $\mathbf{E}(o, t)$ of an object.

When modeling a scene category with an ISM tree, pairs of trajectories are converted into spatial relations. The relations are stored in a table, as was outlined in Section 3.2. A relation topology $\Sigma = (\{o\}, \{\mathbf{R}\})$ describes the same objects and spatial relations as a scene category, but at a different level of abstraction. In a topology, relations are represented on a purely algebraic level instead of explicitly considering their spatial properties as scene categories do. We distinguish the following types of topologies: star topologies $\Sigma_\sigma$, in which a single object $o_F$ (the reference object) is connected to all other objects by one relation each;

complete topologies, in which every object is connected to all other objects; and connected topologies $\Sigma_\nu$, in which each pair of objects is connected by a sequence of relations.

### 3.5. Relation Topology Partitioning—Previous Work

An ISM tree is learned in two steps. In step one, the relations in a scene category are distributed across several ISMs. Step one is covered in this subsection and is part of our previous work ([11]). In step two, the ISMs are then combined into a hierarchical scene classifier. We refer to step two as the tree generation, which is one of the three contributions of this article. It is introduced in the next subsection. Let us assume for step one that a connected relation topology $\Sigma_\nu$ is given for a scene category **S**. Step one distributes the relations in the scene category by partitioning this so-called input topology $\Sigma_\nu$ into a set of star-shaped subtopologies $\{\Sigma_\sigma(j)\}$. The partitioning is performed using a depth-first search that successively selects objects $o_M$ in the topology that are involved in as many relations **R** as possible. We considered each selected object as the center of a star topology $\Sigma_\sigma(j) = (o_M \cup N(o_M), \{\mathbf{R}_M\})$. This star topology also included the relations $\{\mathbf{R}_M\}$ in which the center participated and the neighborhood $N(o_M)$ of the center, i.e., all objects connected to the center by the relations $\{\mathbf{R}_M\}$.

We illustrate how this deep-first search works in Figure 6 using the scene category "Setting-Ready for Breakfast", whose connected relation topology was partitioned into five star topologies in five iterations $j \in \{1, \ldots, 5\}$. Using Video S1 ("Demonstration of object configurations for learning a scene classifier"), we provided footage from the demonstration we recorded for this scene category. The recorded dataset consists of object trajectories that are 112 time steps long. The star topology we extracted first on the left of Figure 6 had "PlateDeep" as its center (each star topology extracted in one iteration is colored green) and all other objects as its neighborhood. We selected the center for the next star topology to be extracted within this neighborhood. We stored the order in which objects $o$ in the input topology $\Sigma_\nu$ would have been chosen as centers for star topologies $\Sigma_\sigma$ in a height function $h_{\{\Sigma_\sigma\}}(o)$. This order would correspond to a breadth-first search. The height function is defined for each object and will be used as a balancing criterion when generating ISM trees in the next subsection, thereby ensuring that the height of the generated tree is minimized. By favoring objects with high degrees, the depth-first search in this subsection ensures that as few star topologies as possible are extracted. All five star topologies extracted from the input topology for "Setting-Ready for Breakfast" can be seen in the leftmost column in Figure 7. Since a depth-first search can completely search any connected graph or relation topology, and its search tree consists of the star topologies we want to extract, we can find a partitioning for any connected input topology.

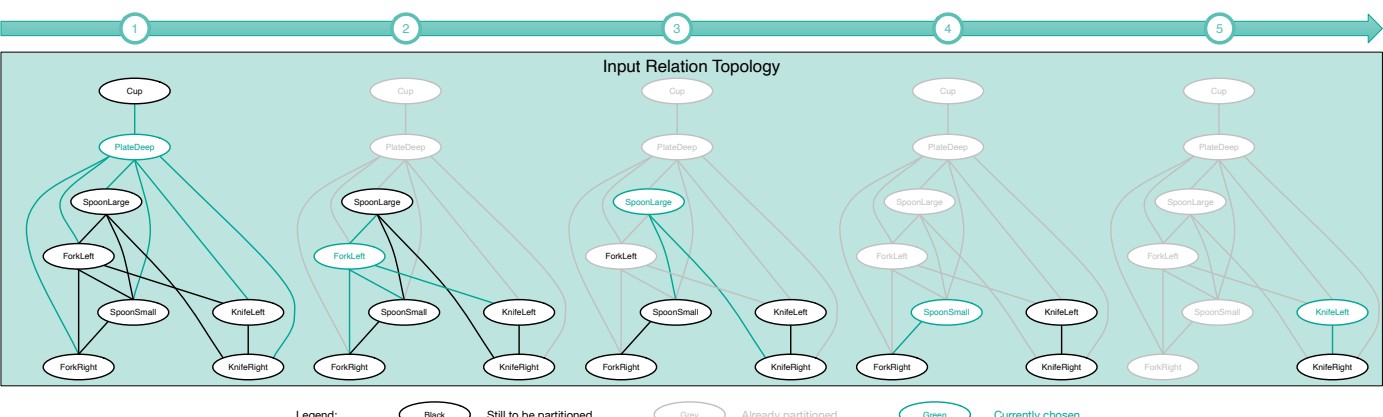

**Figure 6.** Algorithm—how the connected relation topology from which we generated the ISM tree in Figure 8—is first partitioned. The partitioning includes five iterations. It starts at the leftmost graph and ends at the rightmost. In each iteration, a portion of the connected topology, colored green, is converted into a separate star-shaped topology. All stars are on the left of Figure 7.

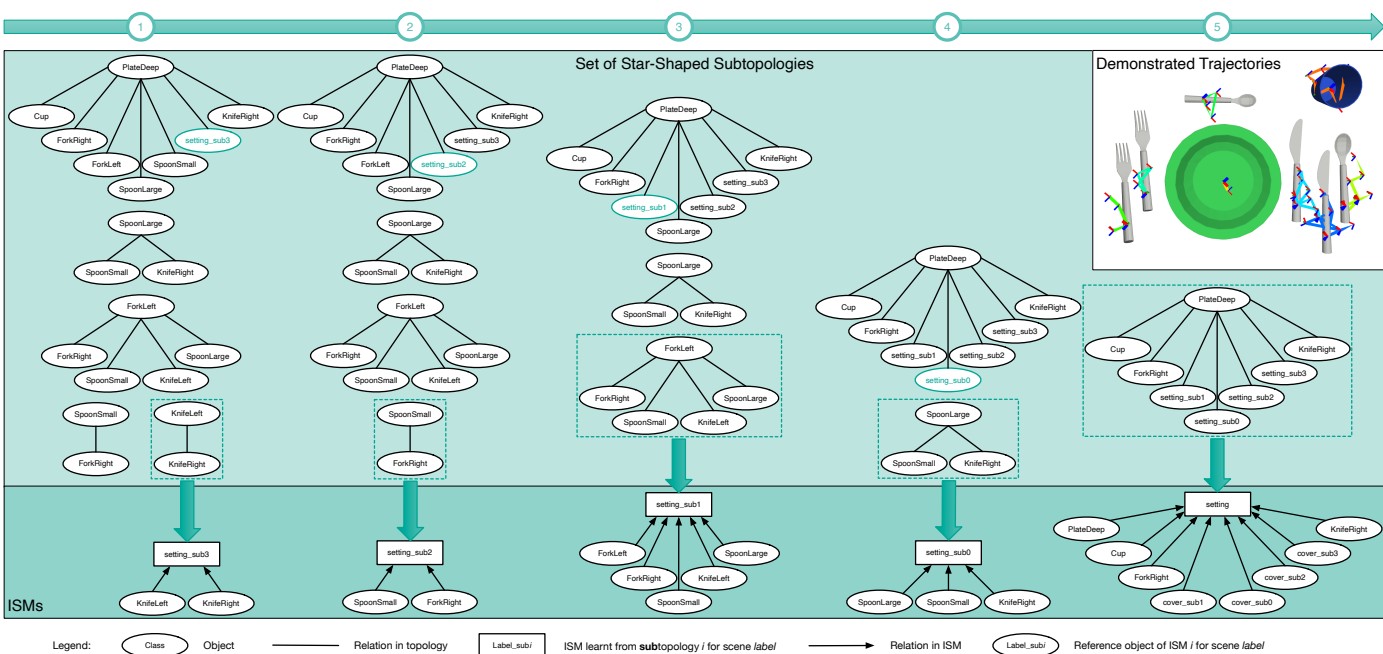

**Figure 7.** Algorithm—how an ISM tree is generated from the star topologies—shown in the leftmost column (see 1). In each of the five depicted iterations, a star topology is selected to learn a single ISM using the object trajectories for scene category "Setting-Ready for Breakfast". Dashed boxes show the selected stars, whereas arrows link these stars to the learned ISMs.

### 3.6. Contribution 1—Generation Algorithm for ISM Trees

Having obtained a set of star topologies in step one, the task in step two is to generate an ISM tree from them. As one of the three contributions of this article, the algorithm we present here models all the extracted star topologies using separate ISMs $m$, which must, however, be linked together to form a tree. Such a tree, generated from the five star topologies shown in the upper left in Figure 7, is visualized as a directed graph in Figure 8. This tree consists of a set $\{m\}$ of five connected ISMs arranged in two levels. At the top of the tree is the root ISM $m_R$, where intermediate results are merged from the four other ISMs $m \in \{m\}$ below. All results we obtain from single ISMs in the tree are hereinafter referred to as recognition results $\mathbf{I}_m$. We use this term to distinguish between the results of single ISMs and the instances $\mathbf{I_S}$ of a scene category $\mathbf{S}$ that result from recognizing scenes with an entire ISM tree.

Within an ISM tree, we also distinguish between real objects found at the leaves $o_L$ and placeholder objects $o_F$ found at the internal vertices of the tree. Leaves and internal vertices are both represented as circles in Figure 8. Internal vertices are named after the scene category and connected to ISMs by green arrows. ISMs are visualized as boxes. As shown in Figure 8, all the inputs for the ISMs at tree level 1 are leaves. Each ISM at this level models relations between real objects and a reference object $o_F$, but compared to Section 3.2, this reference is now a placeholder object in its own right. This placeholder object is used as an interface to pass the recognition results $\mathbf{I}_m$ of an ISM $m$ to another ISM $m' \in \{m\}$ at the next lower level in the tree for further processing. As shown in Figure 8, the reference object "setting_sub1" is used to pass results from the ISM on the lower left to the root ISM. In the root ISM, this reference object is treated as a regular object whose relation to another object is modeled.

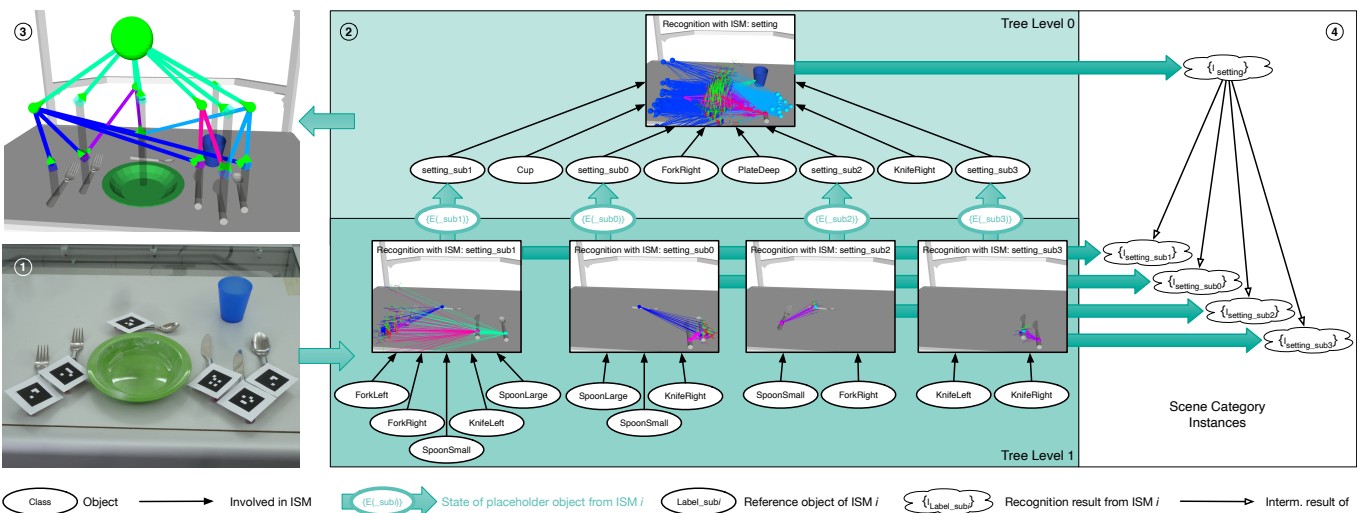

**Figure 8.** Algorithm—how passive scene recognition works with an ISM tree. As soon as object poses estimated from the configuration in **1** are passed to the tree, data flows in **2** from the tree's bottom to its top, eventually yielding the scene category instance shown in **3**. This instance is made up of the recognition results in **4**, which are returned by the single ISMs.

Step two generates ISM trees through two nested loops (the pseudocode is provided in the Appendix A by Algorithm A1). An outer loop converts a star topology into a single ISM in each iteration step, while an inner loop is responsible for attaching the recently generated ISM to the appropriate place in the tree. How this is done for our ongoing example on the scene category "Setting-Ready for Breakfast" can be seen in Figure 7. In this figure, the iterations of the outer loop are visualized column by column from left to right, while the inner loop traverses the star topologies in each column from top to bottom. The order in which the outer loop selects star topologies from the previously extracted set is given by the height function $h_{\{\Sigma_\sigma\}}(o)$ from the previous subsection. This function allows the star topologies $\Sigma_\sigma(j)$ to be processed in the reverse order in which they would have been extracted in a breadth-first search. This ensures that star topologies that could be located in the highest levels of the tree to be generated are attached as close to the root as possible. This minimizes the actual height of the tree. On the left in Figure 7, the last extracted star topology with "KnifeLeft" as its center was accordingly converted into an ISM in the first iteration of the outer loop. The respective selected star topology is surrounded by a dashed rectangle. The conversion was performed using the ISM learning technique from Section 3.2 ([3]). A single ISM *m* was created from this topology and the trajectories $\mathbf{J}(o)$ were demonstrated. Such an ISM is shown in Figure 7 on the left of the lower dark green area.

Before the next iteration of the outer loop can begin, the inner loop still has to answer the question as to which ISM $m'$ the newly created ISM $m$ should be connected. This connection is made utilizing the placeholder reference object $o_F$ of the ISM $m$. Two ISMs $m$ and $m'$ can be connected only if their respective star topologies $\Sigma_\sigma(j)$ and $\Sigma_\sigma(k)$ have an object in common. The connection is created by replacing such a common object in the neighborhood $N(o_M)$ of the center $o_M$ of the latter star topology $\Sigma_\sigma(k)$ with the reference object $o_F$ of the ISM $m$ for the former star topology $\Sigma_\sigma(j)$. To be able to later learn ISM $m'$ with the technique from [3], the trajectory $\mathbf{J}(o)$ demonstrated for the common object is replaced with a placeholder trajectory $\mathbf{J}(o_F)$ for the reference object $o_F$ of the ISM $m$. To minimize the height of the resulting ISM tree, the inner loop starts its search for a star topology $\Sigma_\sigma(k)$ that is suitable for this substitution, which occurs at the topologies that minimize the height function $h_{\{\Sigma_\sigma\}}(o)$ and thus would be located as close to the root as possible. In the leftmost column in Figure 7, the center "KnifeLeft" of the recently selected star topology $\Sigma_\sigma(j)$ is found in the topmost topology $\Sigma_\sigma(k)$, which has "PlateDeep" as its center. In the topmost topology, "KnifeLeft" has been replaced by the reference object

"setting_sub3" of the ISM that was just created. Substitutions by reference objects in star topologies are colored green.

### 3.7. Contribution 2—Recognition Algorithm for ISM Trees

We concretize our definition of scene recognition from Section 3.4 as follows for ISM trees: From an object configuration such as the table setting in **1** in Figure 8, more specifically from the estimated states (the object pose estimation is omitted in Figure 8 for simplicity) $\{\mathbf{E}(o,t)\}$ of the objects, we want to derive instances $\mathbf{I_S}$ of a scene category **S** like the one shown in **3** in Figure 8. Our algorithm for scene recognition using ISM trees is another contribution of this article and involves two steps: An evaluation step, which is exemplified in **2** in Figure 8, and an assembly step, which is exemplified in **4** in Figure 8. Both steps are detailed in this subsection (the pseudocodes for the evaluation and assembly steps are provided in the Appendix A by Algorithms A2–A4). In the evaluation step, all the single ISMs $m$ in a tree are evaluated one by one, i.e., the five ISMs in the example tree in **2**, and all their respective recognition results $\mathbf{I}_m$, are stored for the assembly step. In the assembly step, the recognition results from different ISMs that belong to the same instance of a scene category are combined.

The evaluation step solves two problems: It defines an order in which the ISMs are evaluated and uses an interface to exchange recognition results between ISMs. The actual evaluation of each ISM draws on a technique for classifying scenes with a single ISM. It is from our previous work ([3]) and is outlined in Section 3.2. In an ISM tree, the ISMs cannot all be evaluated simultaneously, since some ISMs $m'$ are supposed to further process the intermediate results of other ISMs $m$. These connections between pairs of ISMs, induced by the reference objects $o_F$, must be taken into account. For example, the evaluation of the root ISM $m_R$ (visualized as a box at tree level 0 in **2** in Figure 8) cannot begin until the evaluation of all four ISMs $m_k$ with $k \in \{0, \dots, 3\}$ at tree level 1 (the dark green area) is completed. By considering these connections, the evaluation step maximizes the efficiency, because each ISM is evaluated exactly once during scene recognition.

The evaluation step begins by sorting the ISMs according to their levels in the tree. This sorted list is traversed using two nested loops such that all the recognition results from the ISMs at tree level $n$ can be stored before the evaluation of the ISMs at tree level $n-1$ begins. In **2** in Figure 8, this equates to evaluating the ISMs from bottom to top line by line. If only real objects, i.e., only leaves $o_L$ and no internal vertices, are involved in the evaluation of the ISMs at a certain level, it is sufficient that the evaluation step distributes the different object states $\{\mathbf{E}(o,t)\}$ that describe the object configuration to the appropriate ISMs. For instance, at tree level 1 in **2**, this is the case. If internal vertices are involved in an ISM, reference objects $o_F$ or, more precisely, their placeholder states $\mathbf{E}(o_F)$ should be computed before the ISM's evaluation. For instance, to evaluate the root ISM $m_R$ at level 0 in **2**, such placeholder states should be derived for the reference objects "setting_sub$k$" from all the recognition results returned by the ISMs $m_k$ and passed to the root. These placeholder states are visualized as vertical green arrows emanating from the ISMs from which they originate and pointing to the internal vertex where they are further processed. Each placeholder state includes a pose $\mathbf{T}_F$, which is the pose of a recognition result returned by an ISM. Such a pose is the location in the ISM's accumulator at which the recognition result (a highly rated combination of votes) has been identified during the evaluation of the ISM.

Each ISM $m$ that is not the root ISM may pass zero to a multitude of recognition results to another ISM $m'$ in the tree. When the ISM $m'$ is evaluated, each of these results is considered as a separate input, which yields more recognition results in this ISM. These results should be passed on to a third ISM. We implemented two strategies that mitigate this effect to avoid a combinatorial explosion in scene recognition: Firstly, when we generate ISM trees; the height function $h_{\{\Sigma_\sigma\}}(o)$ is used to minimize tree heights and, thus, the lengths of the chains of interdependent ISMs in a tree. Secondly, the number of placeholder

states $\mathbf{E}(o_F)$ emanating from each ISM is limited by discarding all the recognition results that have been assigned too low confidence levels $b(o_F)$.

The evaluation step ends once it has evaluated the ISM at the root. The recognition results from the different ISMs are visualized as clouds in **4** in Figure 8. The results are connected through horizontal green arrows to those ISMs where they were computed. The task of the assembly step that now begins is to determine across ISMs which recognition results belong to the same instance $\mathbf{I_S}$ of a scene category and to assemble such instances. As in **4**, the assembly step starts at the results of the root. It recursively compares from top to bottom the stored recognition results $\mathbf{I}_m$, $\mathbf{I}_{m'}$ according to the connections between pairs of ISMs: $m$ and $m'$. Such a recursion chain is started in **4** for each recognition result $\mathbf{I}_{\text{setting}}$ computed by the root ISM $m_R$. During each recursion chain, a recognition result $\mathbf{I}_{\text{setting}}$ is compared with the intermediate results $\mathbf{I}_{\text{setting\_sub}k}$ of the different ISMs $m_k$ at level 1. For a comparison to assign two recognition results, $\mathbf{I}_m$ and $\mathbf{I}_{m'}$, to the same instance, two conditions must be met: Firstly, these results must come from two ISMs, $m$ and $m'$, that exchanged reference objects $o_F$. Secondly, the very same reference object must have been involved in both recognition results. The second condition is satisfied if one of the reference objects in each of the two recognition results, $\mathbf{I}_m$ and $\mathbf{I}_{m'}$, has the same state $\mathbf{E}(o_F)$.

### 3.8. Relation Topology Selection—Previous Work

While we explained how we partition relation topologies in Section 3.5, we did not address how to determine the relation topology to partition. Our novel generation algorithm can derive an ISM tree for any kind of connected topology, but not every topology is equally suitable for learning a classifier. Figure 9 illustrates how omitting the wrong relations can lead to recognition errors, i.e., false positive results. We define a scene category instance $\mathbf{I_S}$ to be a false positive if the scene recognition assigns it a confidence level $b(\mathbf{I_S})$ that exceeds a given threshold, whereas its underlying object configuration $\{o'\}$ does not sufficiently match that scene category. Image **1** in Figure 9 visualizes a result of the scene recognition for the "Setting-Ready for Breakfast" category. To generate the tree employed here, we used a star topology whose center is the green plate.

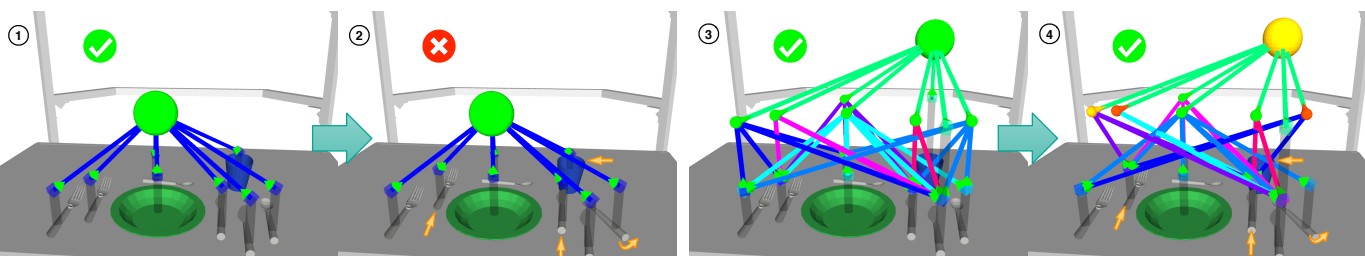

**Figure 9.** Results from ISM trees for the same demonstration, but learned with different topologies. Valid results are annotated with a tick in green, invalid results by a cross in red. In images **1** and **2**, a star is used. In images **3** and **4**, a complete topology is used. In images **1** and **3**, valid object configurations are processed. Instead, images **2** and **4** show invalid configurations.

From a valid place setting (as in **1** and **3** in Figure 9), we expect that utensils such as forks, knives, and spoons be on the "correct" sides of the plate. In addition to this first set of rules, others require that forks, knives, and spoons be oriented parallel to each other. There are also rules regarding the relative distances of utensils from the edge of the table. If a star topology is used to cover the first set of rules, the other rules cannot be modeled with this topology. For this reason, the ISM tree created from a star topology already used in **1** in Figure 9 returned a false positive in **2**. The invalid configuration $\{o'\}$ shown in **2** differs from the valid $\{o\}$ in **1** in that several relative poses between object pairs that do not involve the plate are invalid. The ISM tree, however, did not notice these differences visualized by yellow arrows in **2**. This false positive is indicated by a white cross on a red background, whereas white check marks on a green background indicate true positives.

To prevent false positives, ISM trees could instead be learned from all $n \cdot (n-1)/2$ spatial relations that can be defined for a set of $n$ objects, i.e., from a complete relation topology. The fact that ISM trees from such complete topologies do not yield false positives is illustrated in **3** and **4** in Figure 9. In **3** and **4**, such a tree has been applied to the object configurations $\{o\}$ and $\{o'\}$ from **1** and **2** in Figure 9. The result in **4** is not a false positive, as some of the ISMs in the tree recognized that some of the relations modeled by them were not fulfilled. The colors of the spheres above the single ISMs in the tree indicate to which degree their respective relations are fulfilled. However, a disadvantage of complete topologies is the excessive number of relations that must be checked during scene recognition. In general, the cost of scene recognition with ISM trees is closely related to the number of relations represented. The fact that recognition with complete topologies is generally intractable has also been reported ([27]) for other part-based models.

The question arose as to how to find a connected topology that is different from the edge cases, which are the star and complete topologies, as a middle ground. Such a topology would yield an ISM tree that combines efficiency and representational power. To identify such a relation topology most generically, we used two domain-unspecific goodness measures in our previous work ([11]): the false positive rate numFPs() of the scene recognition and the average time consumption avgDur() of the scene recognition. Based on these measures, we formalized the selection of the relation topologies as a combinatorial optimization problem. The challenge in this selection is the exponential number $2^{n \cdot (n-1)/2}$ of relation topologies that can be defined for $n$ objects. Given the number of topologies among which to choose, we used a local search technique to develop a relation topology selection procedure. Its basic idea was to iteratively adjust a relation topology by adding, removing, or exchanging relations until a topology was found that contained only those relations that were most important for recognizing a scene category. The result, a so-called optimized topology, was then used to learn an ISM tree from it.

## 4. Methods—Active Scene Recognition

### 4.1. State Machine and Next-Best-View Optimization—Previous Work

In the previous section on passive scene recognition (PSR), we ignored the question regarding under which conditions object pose estimation can obtain "object estimates" for scene recognition. Our approach to creating suitable conditions in spatially distributed and cluttered indoor environments is to have a mobile robot adopt camera views from which it can perceive searched objects. To this end, in two previous works ([7,8]), we introduced active scene recognition (ASR)—an approach that connects PSR with a three-dimensional object search within a decision-making system. We implemented ASR as a state machine consisting of two search modes (states), DIRECT_SEARCH and INDIRECT_SEARCH, that alternate. We then integrated this state machine with the MILD robot shown in Figure 1 so that ASR could decide on the presence of $n$ scene categories in the environment visible in Figure 2.

ASR starts in DIRECT_SEARCH mode, which is tasked with acquiring initial object estimates. For this purpose, we developed two strategies to identify suitable camera views and to move them. The first ("informed") strategy is based on prior knowledge about possible placements of objects, e.g, from demonstrations of scene categories. If this informed search does not yield object estimates, an uninformed strategy ([6]) is used to explore the entire environment uniformly. As soon as at least one object estimate is obtained, the direct search stops, and the INDIRECT_SEARCH mode starts instead.

The other mode INDIRECT_SEARCH consists of a loop in which three substates (passive scene recognition, a technique for predicting the poses of searched objects, and a 3D object search) alternate. The loop starts in the first substate SCENE_RECOGNITION, in which PSR is performed with ISM trees on the currently available object estimates. The results of SCENE_RECOGNITION are instances of scene categories. Some instances may not contain all the objects belonging to their category. Therefore, it is the task of the other two substates in the loop to complete such partial instances. The second substate

OBJECT_POSE_PREDICTION uses ISM trees to predict locations of objects that would allow for the completion of these instances. When using ISM trees, some object poses may need to be predicted using entire sequences of spatial relations. This is prone to a combinatorial explosion: An algorithm presented in our previous work ([7]) for predicting poses suffered from such an explosion. In the next subsection, we address this problem with an efficient algorithmic solution (one of the contributions of this article).

The third substate of the loop, RELATION_BASED_SEARCH, uses predicted object poses to search for these objects in 3D, i.e., to determine camera views that are promising for finding them. Whenever such a view has been determined, the robot moves there and tries to localize objects in 6-DoF. In [8], we formalized finding suitable camera views as a next-best-view (NBV) optimization problem. The algorithm with which we addressed this problem had to search for a camera view that maximized an objective function, thus starting from predicted object poses and the current robot pose. This objective function modeled the success probability of the object localization, as well as the time required to reach the view and perform localization. Our approach allowed for both optimizing the views and deciding which objects to search in them.

### 4.2. Contribution 3—Object Pose Prediction Algorithm

Our approach to predicting the poses of searched objects with the help of ISM trees is similar to an inversion of the scene recognition. Scene recognition infers from the known states of objects which instances of a scene category $\mathbf{S}$ the states correspond to. On the contrary, object pose prediction infers hypotheses about the possible poses $\mathbf{T}_P$ of the missing objects $o_P$ from a known scene category instance $\mathbf{I_S}$ and its location $\mathbf{T}_F$. Since these predicted poses must be suitable for a 3D object search, both the 3-DoF positions of the missing objects and their 3-DoF orientations must be predicted. Knowledge about the expected orientation of a searched object can determine the success or failure of the object localization. The poses predicted by the algorithm presented in this subsection are visualized as coordinate systems in **2** and **4** in Figure 10. ISM trees allow us to infer object poses from spatial relations $\mathbf{R}$, i.e., depending on the known poses $\mathbf{T}$ of already found objects $o$. The flexibility of this approach is illustrated in Figure 10 for the scene category "Setting-Ready for Breakfast": If an incomplete instance of this scene category is rotated as between **1** and **3**, the object poses predicted from them in **2** and **4** rotate with it without the need for adjustments.

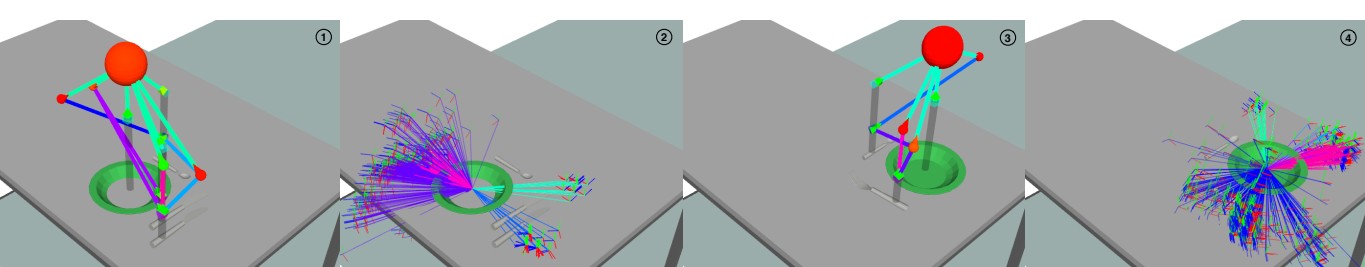

**Figure 10.** Example of how scene recognition (see images **1** and **3**) and object pose prediction (see images **2** and **4**) can compensate for changing object poses. The results they return are equivalent, even though the localized objects are rotated by 90° between images **1** and **2** and between images **3** and **4**.

Predicting object poses with ISM trees consists of two steps and is the third and final contribution of this article: In step one, precomputations are performed to identify those parts of an ISM tree that provide a fast and reliable prediction. In step two, these precomputations are used to predict the poses of the searched objects. Figure 11 refers to the ISM tree that models the scene category "setting", and it has already been used in Figure 8. Since some objects from the scene category are involved in multiple relations, several leaves $o_L$ in the tree correspond to the same object. In this way, "ForkRight" is

represented at both levels of the tree. To predict object poses using the leaf for "ForkRight" at tree level 1, one would have to combine the spatial relations from the ISMs "setting" and "setting_sub1". On level 0, a single relation in the ISM "setting" is sufficient. Since the accuracy of the predicted poses depends on the number of relations used, step one precomputes the shortest sequences of ISMs between any object $o$ in a scene category **S** and the root $m_R$ of the tree. All nontrivial sequences are defined as paths $\mathbf{P}(m_R, o)$ consisting of $l$ pairs $(m_k, m_{k+1})$ of connected ISMs (see Equation (1)). Paths end at the ISM $m_{l+1}$ that contains the appropriate leaf $o_L$ for predicting the pose of object $o$. We compute the shortest paths using a breadth-first search that traverses trees, as is shown in Figure 11, from top to bottom.

$$\mathbf{P}(m_R, o) = \{(m_k, m_{k+1}) |\ m_1 = m_R \wedge \forall k : m_k \text{ connected with} \\ m_{k+1} \text{ by an } o_F \wedge o \text{ is a } o_L \text{ of } m_{l+1}\} \tag{1}$$

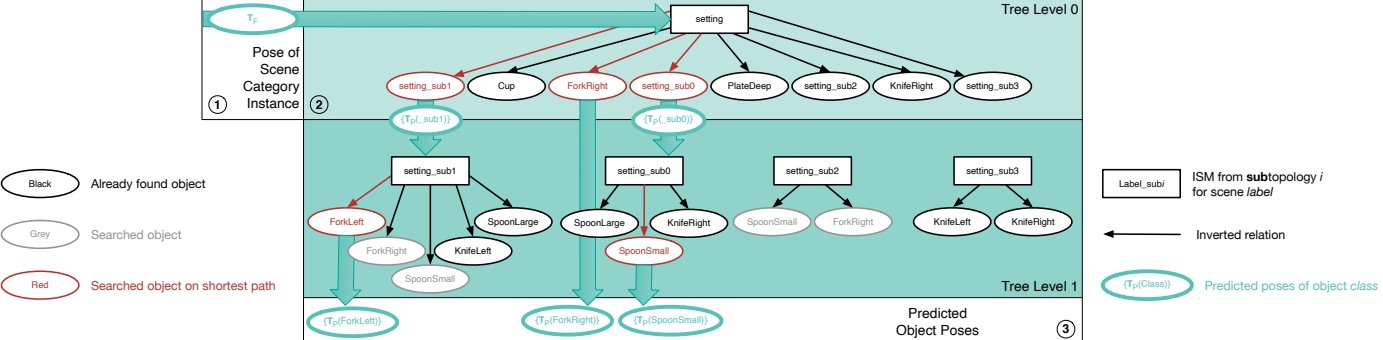

**Figure 11.** Algorithm—how the poses of searched objects are predicted with an ISM tree. The incomplete scene category instance in **1** is input and causes data to flow through the tree in **2**. Unlike scene recognition, data flows from the tree's top to its bottom, while relative poses from different inverted spatial relations are combined. The resulting predictions are visible in **3**.

Step two, the actual pose prediction algorithm, derives possible poses for the searched objects from these paths and a partial scene category instance $\mathbf{I_S}$. This is performed via three nested loops: The innermost loop (the pseudocode for the innermost loop is provided in the Appendix A by Algorithm A5) computes exactly one pose estimate $\mathbf{T}_P$ per searched object $o_P$. Two outer loops (the pseudocode for the outer loops is provided in the Appendix A by Algorithm A6) call this innermost loop until a specified number $n_P$ of poses is predicted for each searched object. Figure 11 shows how the innermost loop operates on an ISM tree. First, as shown by the horizontal green arrow in **1**, it passes the pose $\mathbf{T}_F$ of instance $\mathbf{I_S}$ to the root ISM. Starting from the root ISM $m_R$ in **2**, it evaluates all of the ISMs for the shortest path $\mathbf{P}(m_R, o_P)$ to a suitable leaf. For "ForkLeft", such a leaf is located on the far left of tree level 1. A predicted pose is visualized in **3** as a green circle and connected by a green arrow to the leaf from which it results.

The algorithm originating from our previous work ([7]) was unable to efficiently predict object poses because it processed all of the relative poses that make up a spatial relation in an ISM. Across multiple ISMs, this would lead to a combinatorial explosion: If it generated a prediction for each relative pose in a relation of an ISM $m_k$ at tree level $k$ and passed the prediction as a possible pose of a reference object to another ISM $m_{k+1}$ at level $k + 1$, each of these would be combined with all the relative poses in a relation of ISM $m_{k+1}$. Also, the algorithm did not use the shortest paths. Combinatorial explosion is avoided in the new method presented here by processing one random relative pose per relation instead of all the poses. More precisely, the innermost loop selects one relative pose $\mathbf{T}_{jk}$ from each ISM along the shortest path and inverts all such poses. To visualize in Figure 11 how the spatial relations have been inverted to predict object poses, all arrows point from top to bottom instead of bottom to top. The shortest paths used to predict poses are colored red. The pose $\mathbf{T}_F$ of the incomplete instance is multiplied by all these sampled and inverted

relative poses $\mathbf{T}_{kj}$ so that one of the sought pose hypotheses $\mathbf{T}_P$ is obtained. For instance, to predict an absolute pose of "ForkLeft", the innermost loop randomly selects one relative pose from the ISMs "setting" and "setting_sub1", respectively.

### 4.3. Discussion of Methods

Sections 3 and 4 have closed three algorithmic gaps at the core of active scene recognition (ASR)—our implicit shape model trees—that previously made ASR impractical. It was not known how to generate ISM trees, nor did fellow researchers know how to use them for scene recognition. Moreover, the previously suggested technique for pose prediction was too inefficient to be used on a physical mobile robot. The two abovementioned sections resolved these issues by answering the following research questions: What do data and control flow in an ISM tree look like during scene recognition? How must ISMs be combined into a tree to represent any connected relation topology? In which manner should relations in an ISM tree be evaluated to obtain pose predictions efficiently enough to make ASR possible?

## 5. Experiments and Results

### 5.1. Overview

We present experiments for PSR with ISM trees in Section 5.2 and for our approach to ASR in Section 5.3. Except for explicitly labeled experiments in Sections 5.2.5, 5.3.4, and 5.3.5, our PSR and ASR approaches have been evaluated exclusively on measurements from physical sensors. The input for these real-world experiments was acquired by the pivoting sensor head of our MILD mobile robot shown in Figure 1. Our approach to ASR controlled both the sensors and actuators of this physical robot. The robot operated in our experimental setup, which mimicked some aspects of a kitchen (see Figure 2). Our approaches to PSR and ASR were run on a PC with an Intel Xeon E5-1650 v3 3.50 GHz CPU and 32GB DDR4 RAM. In Sections 5.3.1–5.3.3, ISM trees for ten scene categories were used to evaluate our ASR approach. In each subsection, ASR is expected to provide estimates for all the existing scenes. In addition to these ASR experiments on a physical robot, we performed an experiment in simulation in Section 5.3.4 to compare the time consumption of our approach to ASR with two alternatives.

### 5.2. Evaluation of Passive Scene Recognition

#### 5.2.1. Scene Category "Office"

In this experiment, we evaluated how well our scene recognition approach captures the properties of spatial relations throughout an ISM tree. We did this by investigating how changes in individual object poses affect the recognized scene category instances. The scene category used is named "Office" and consists of four objects with fiducial markers attached to them to maximize object localization accuracy: Mouse, Keyboard, LeftScreen, and RightScreen. Video S2 ("Demonstration of the scene category: Office") shows how we demonstrated this category. Image **1** in Figure 12 visualizes 1 of the 51 object configurations included in the dataset for this demonstration. The relative poses that make up the spatial relations of the learned ISM tree are visualized in **3** in Figure 12. The demonstration includes two relative movements between object pairs. The first relative movement involves both screens. It creates a relation that consists of nearly identical relative poses, as depicted in the middle of **3**. The second relative movement between Mouse and Keyboard creates a much more variable spatial relation. The tree learned for "Office" consists of one ISM labeled "Office" and another labeled "Office_sub0".

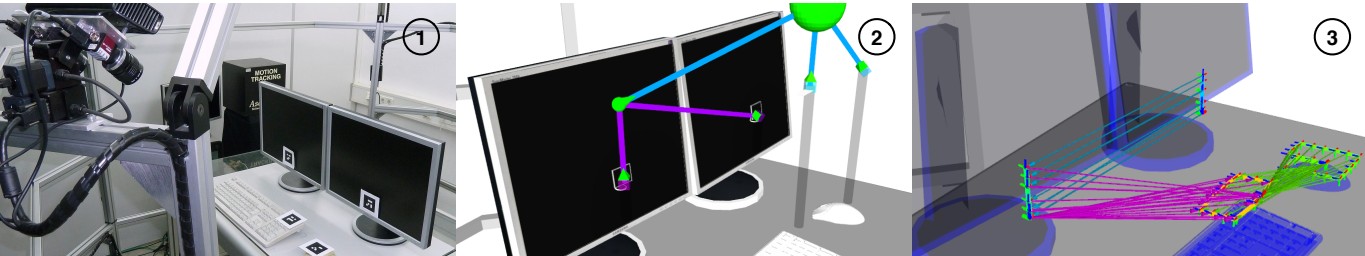

**Figure 12.** Image **1**: Snapshot of the demonstration for scene category "Office". Image **3**: The ISM tree for this category, including the relative poses in its relations and the demonstrated poses. Image **2**: Result of applying this tree onto the configuration "Correct-configuration".

5.2.2. Parameters Describing Results of Scene Recognition

Scene recognition was performed on nine object configurations to analyze what impact changing the pose of an object has on the parameters in scene category instances. Since scene recognition is deterministic, each configuration was processed only once. In each configuration, an object pose either differs in its position or its orientation from those expected by the spatial relations in the ISM tree. Tables 1 and 2 show how the ISMs "Office_sub0" and "Office" quantified these differences. In both tables, color coding indicates the appropriateness of the values they contain. Green stands for results we consider excellent, yellow stands for good results, and red indicates problems.

**Table 1.** We specify position and orientation compliances, as well as a similarity rating for all measured object poses, which ISM "Office_sub0" processed. The value of the recognition result's objective function (returned by the ISM) is also provided. See Section 5.2.2 for definitions of the objective function, rating, and compliances.

| Object Configuration | LeftScreen | | | RightScreen | | | Obj. Function |
|---|---|---|---|---|---|---|---|
| | Simil. | Pos. | Orient. | Simil. | Pos. | Orient. | |
| Correct configuration | 1.00 | 1.00 | 1.00 | 1.00 | 1.00 | 1.00 | 2.00 |
| RightScreen half-lowered | 1.00 | 1.00 | 1.00 | 0.72 | 0.72 | 1.00 | 1.72 |
| RightScreen fully lowered | 0.00 | 0.00 | 0.00 | 1.00 | 1.00 | 1.00 | 1.00 |
| LeftScreen half front | 1.00 | 1.00 | 1.00 | 0.60 | 0.61 | 0.98 | 1.60 |
| RightScreen half right | 1.00 | 1.00 | 1.00 | 0.96 | 0.97 | 0.99 | 1.96 |
| LeftScreen half-rotated | 0.54 | 0.93 | 0.58 | 1.00 | 1.00 | 1.00 | 1.54 |
| LeftScreen fully rotated | 0.00 | 0.00 | 0.00 | 1.00 | 1.00 | 1.00 | 1.00 |
| Mouse half right | 1.00 | 1.00 | 1.00 | 0.99 | 1.00 | 0.99 | 1.99 |
| Mouse half-rotated | 1.00 | 1.00 | 1.00 | 0.99 | 0.99 | 1.00 | 1.99 |

Each estimated scene category instance in Figure 13 can also be represented as a set of parameters whose values can be found in the same row of the two tables. For each object, two compliance parameters express the degree to which its estimated position and orientation complied with a spatial relation in which the object was involved. Formal definitions of these compliances can be found in [9]. The compliances were normalized to $[0, 1]$, where one expresses a perfect match, and zero represents a lower bound below which objects are excluded from scene category instances. A similarity measure was derived for each object by multiplying both compliances. Adding up all the similarity measures in a table row yields the value of the objective function for an ISM $m$, which is the nonnormalized equivalent to its confidence level. This value describes the extent to which all of these objects, either directly involved with the ISM $m$ or involved with another ISM $m'$ to which $m$ is related, contributed to the recognition result produced by ISM $m$.

**Table 2.** This table shows the same quantities as Table 1, but this time for all measured object poses, which ISM "Office" processed. The same applies to the recognition result's objective function.

| Object Configuration | Keyboard | | | Mouse | | | Office_sub0 | | | Obj. Function |
|---|---|---|---|---|---|---|---|---|---|---|
| | Simil. | Pos. | Orient. | Simil. | Pos. | Orient. | Simil. | Pos. | Orient. | |
| Correct configuration | 0.98 | 0.98 | 1.00 | 1.00 | 1.00 | 1.00 | 2.00 | 1.00 | 1.00 | 3.98 |
| RightScreen half-lowered | 0.98 | 0.98 | 1.00 | 1.00 | 1.00 | 1.00 | 1.72 | 1.00 | 1.00 | 3.70 |
| RightScreen fully lowered | 1.00 | 1.00 | 1.00 | 1.00 | 1.00 | 1.00 | 1.00 | 1.00 | 1.00 | 3.00 |
| LeftScreen half front | 0.97 | 0.98 | 0.99 | 1.00 | 1.00 | 1.00 | 1.47 | 0.89 | 0.98 | 3.43 |
| RightScreen half right | 0.99 | 0.99 | 1.00 | 0.99 | 0.99 | 1.00 | 1.96 | 1.00 | 1.00 | 3.94 |
| LeftScreen half-rotated | 0.98 | 0.98 | 1.00 | 1.00 | 1.00 | 1.00 | 1.46 | 0.93 | 1.00 | 3.45 |
| LeftScreen fully rotated | 0.93 | 0.95 | 0.97 | 0.95 | 0.98 | 0.96 | 1.00 | 1.00 | 1.00 | 2.87 |
| Mouse half right | 0.99 | 0.99 | 1.00 | 0.86 | 0.86 | 1.00 | 1.99 | 1.00 | 1.00 | 3.83 |
| Mouse half-rotated | 1.00 | 1.00 | 1.00 | 0.77 | 1.00 | 0.78 | 1.96 | 0.98 | 0.99 | 3.73 |

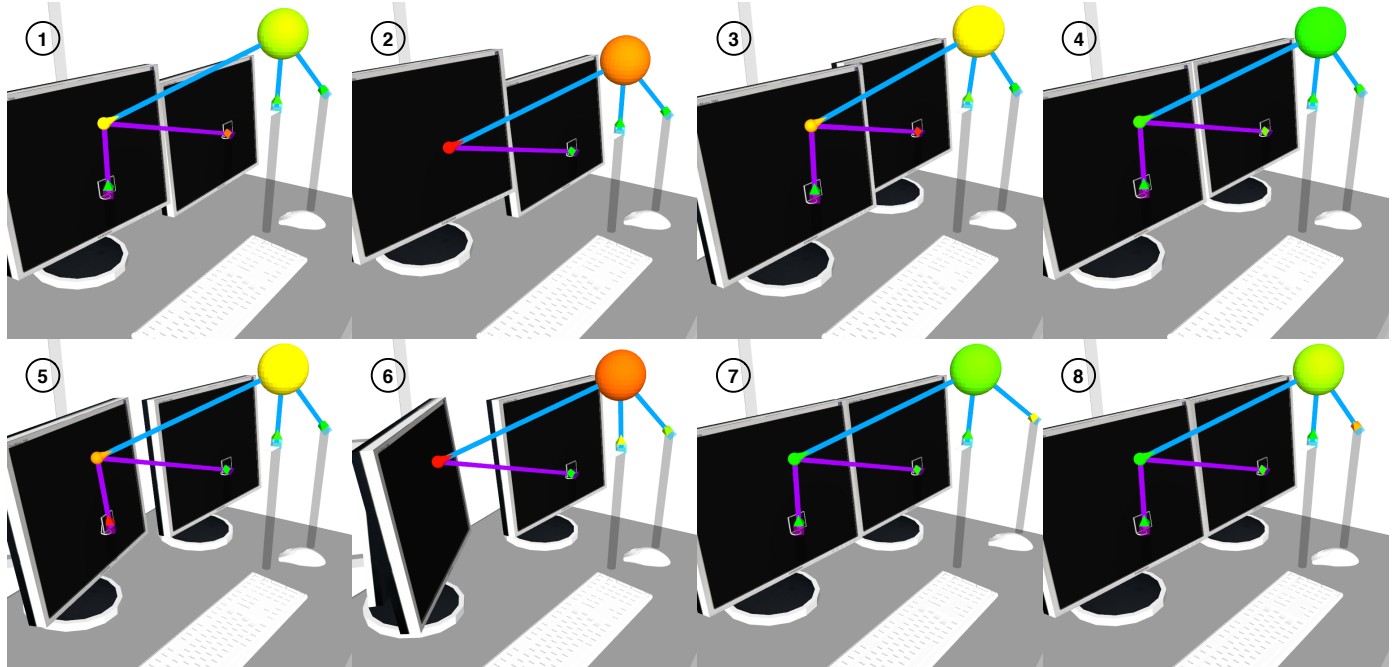

**Figure 13.** Visualization of the scene category instances that an ISM tree for the category "Office" recognized in the physical object configurations "RightScreen-half-lowered" (image **1**), "RightScreen-fully-lowered" (image **2**), "LeftScreen-half-front" (image **3**), "RightScreen-half-right" (image **4**), "LeftScreen-half-rotated" (image **5**), "LeftScreen-fully-rotated" (image **6**), "Mouse-half-right" (image **7**), and "Mouse-half-rotated" (image **8**).

5.2.3. Influence of Object Poses on Passive Scene Recognition

As seen in the uppermost lines of Tables 1 and 2, all the compliances concerning positions and orientations were close to one. Thus, the objective function reached its maximum. The values of the objective function correspond to the number of objects that each ISM considered. In "RightScreen-half-lowered", RightScreen was moved downwards by 0.05 m, as can be seen in **1** in Figure 13. The compliances for "RightScreen-half-lowered" in Table 1 validate that ISM "Office_sub0" correctly noticed that RightScreen had been slided, but not rotated. To increase the discrepancy between the positions of LeftScreen and RightScreen, LeftScreen was displaced further upwards by 0.035 m in "RightScreen-fully-lowered". In contrast, the nonzero compliances in Table 1 indicate that no object in "RightScreen-half-lowered" had been excluded from the scene category instance as

we intended; this was different in "RightScreen-fully-lowered". The positional difference between the screens was sufficient to exclude one screen. However, the yellow coloring of the corresponding compliances in Table 1 makes it clear that it is suboptimal that ISM "Office_sub0" excluded the less-displaced LeftScreen. ISM trees are more sensitive to displacements of the reference objects of their ISMs than to those of nonreference objects.

When shifting LeftScreen forward by 0.05 m in "LeftScreen-half-front", instead of moving RightScreen, the scene recognition reveals that the displacements were always considered from the perspective of the reference object. The two aforementioned phenomena, although counterintuitive, did not affect the values calculated for the overall objective function and could be systematically compensated. In configuration "RightScreen-half-right", we displaced RightScreen this time. We moved this screen to the right so that our experiments covered all the directions in 3D space where shifting is possible. When being moved to the right by 0.015 m, RightScreen was displaced less than in "RightScreen-half-lowered". The objective function values in Table 1 show that ISMs can be sensitive enough to notice such slight differences.

After checking whether the ISM trees detect the translations of objects, the same should be done for rotations. We rotated LeftScreen by 15° in "LeftScreen-half-rotated" and by 30° in "LeftScreen-fully-rotated". Comparing the configurations in which LeftScreen was rotated with those in which RightScreen was lowered reveals that the rotations were just as precisely detected as translations. After having analyzed how changing object poses affects ISM "Office_sub0", the next configurations are to show that changes are treated equally in the root ISM "Office". In the configuration "Mouse-half-right", Mouse was pushed 0.11 m to the right. This represents a displacement larger than those of both screens together in "RightScreen-fully-lowered". However, Mouse was not excluded from the corresponding scene category instance in **7** in Figure 13. This shows that we can control how permissive spatial relations are through the demonstrations we record. In "Mouse-half-rotated", Mouse was rotated by 15° instead of being shifted. The fact that the objective function of ISM "Office" returned the same value for "Mouse-half-rotated" and "LeftScreen-half-rotated" proves that we can further influence whether scene recognition is permissive concerning positions or orientations. Overall, these experiments confirm that ISM trees can identify whether an object has been shifted or rotated in various directions. They can also estimate the sizes of such displacements.

### 5.2.4. Scene Categories Demonstrated for ASR Evaluation

This subsection is devoted to the scene categories that we demonstrated to evaluate ASR and that are named in Table 3. The next subsection is then devoted to the performance of the ISM trees learned from the demonstrations we recorded for these categories. The demonstrations and the evaluation of ASR took place in the kitchen setup depicted in Figure 2 or **1** in Figure 14. Object configurations were demonstrated in areas of the setup such as the cupboard at the top of **1**, the shelves on its right, and the tables. The cupboard and shelves were filled with clutter. We recorded all the object poses with the camera head of our MILD robot. Markers were only used on the cutlery to compensate for reflections. The ISM trees for all the scene categories in Table 3 resulted from topologies optimized using the relation topology selection (RTS), which we outlined in Section 3.8. This table contains the durations (lengths) of the object trajectories, as well as the numbers of objects in the datasets of each scene category. Some of the categories are visualized in Figure 14. Whereas **1**, **2**, **6**, **7**, and **8** show the object trajectories and spatial relations, **3**, **4**, and **5** show the snapshots of demonstrations. As ISM trees are generative models, different scene categories can contain the same objects and model similar relations. This also allows for searching for different scenes at the same time.

In the different areas of our setup, the objects can be arranged horizontally or vertically in 2D. However, we defined scene categories that span multiple areas and thus extended into 3D. For instance, the category "Cereals-on Shelf" in **5**, **6**, and **8** relates parts of a table setting to the food and drinks stored on the shelves. Food, drinks, and the shelves are also

part of "Drinks-on Shelf" in **2** and "Sandwich-on Shelf". The ISM trees for these scene categories contained relations of a considerable length, such as those drawn in **2**. The object configurations corresponding to these scene categories are truly three-dimensional, as they extend both horizontally and vertically. A close-up view of the vertical relations in "Cereals-on Shelf" is provided in **6**, whereas the horizontal ones are shown in **8**. Except for the shelves, the three categories "Sandwich-Setting" in **1** and **3**, "Cereals-Setting" in **4** and **7**, and "Drinks-Setting" consist of the same objects as their "...-on Shelf" counterparts. The former three expect food and drinks to be located on a table, not on the shelves.

**Table 3.** Performance of ISM trees learned for scene categories in our kitchen and optimized by relation topology selection. Color coding as in Table 1.

| Scene Category | Trajectory Length | # Objects | Relations | | | numFPs() [%] | | | avgDur() [s] | | |
|---|---|---|---|---|---|---|---|---|---|---|---|
| | | | Star | Optim. | Complete | Star | Optim. | Complete | Star | Optim. | Complete |
| Setting—Ready for Breakfast | 112 | 8 | 7 | 15 | 28 | 38.56 | 3.86 | 0 | 0.044 | 1.228 | 7.558 |
| Setting—Clear the Table | 220 | 8 | 7 | 15 | 28 | 19.59 | 2.92 | 0 | 0.196 | 2.430 | 8.772 |
| Cupboard—Filled | 103 | 9 | 8 | 15 | 36 | 23.70 | 8.09 | 0 | 0.020 | 0.140 | 1.070 |
| Dishwasher Basket—Filled | 117 | 10 | 9 | 9 | 45 | 15.38 | 0 | 0 | 0.053 | 0.060 | 1.099 |
| Sandwich—Setting | 92 | 5 | 4 | 4 | 10 | 0 | 0 | 0 | 0.019 | 0.019 | 0.131 |
| Sandwich—On Shelf | 98 | 6 | 5 | 7 | 15 | 29.05 | 6.70 | 0 | 0.017 | 0.042 | 0.264 |
| Drinks—Setting | 23 | 3 | 2 | 2 | 3 | 67.02 | 67.02 | 0 | 0.001 | 0.001 | 0.004 |
| Drinks—On Shelf | 44 | 4 | 3 | 3 | 6 | 44.51 | 11.54 | 0 | 0.003 | 0.008 | 0.013 |
| Cereals—Setting | 52 | 4 | 3 | 3 | 6 | 33.67 | 0 | 0 | 0.002 | 0.004 | 0.012 |
| Cereals—On Shelf | 98 | 5 | 4 | 7 | 10 | 9.50 | 1.68 | 0 | 0.014 | 0.053 | 0.181 |

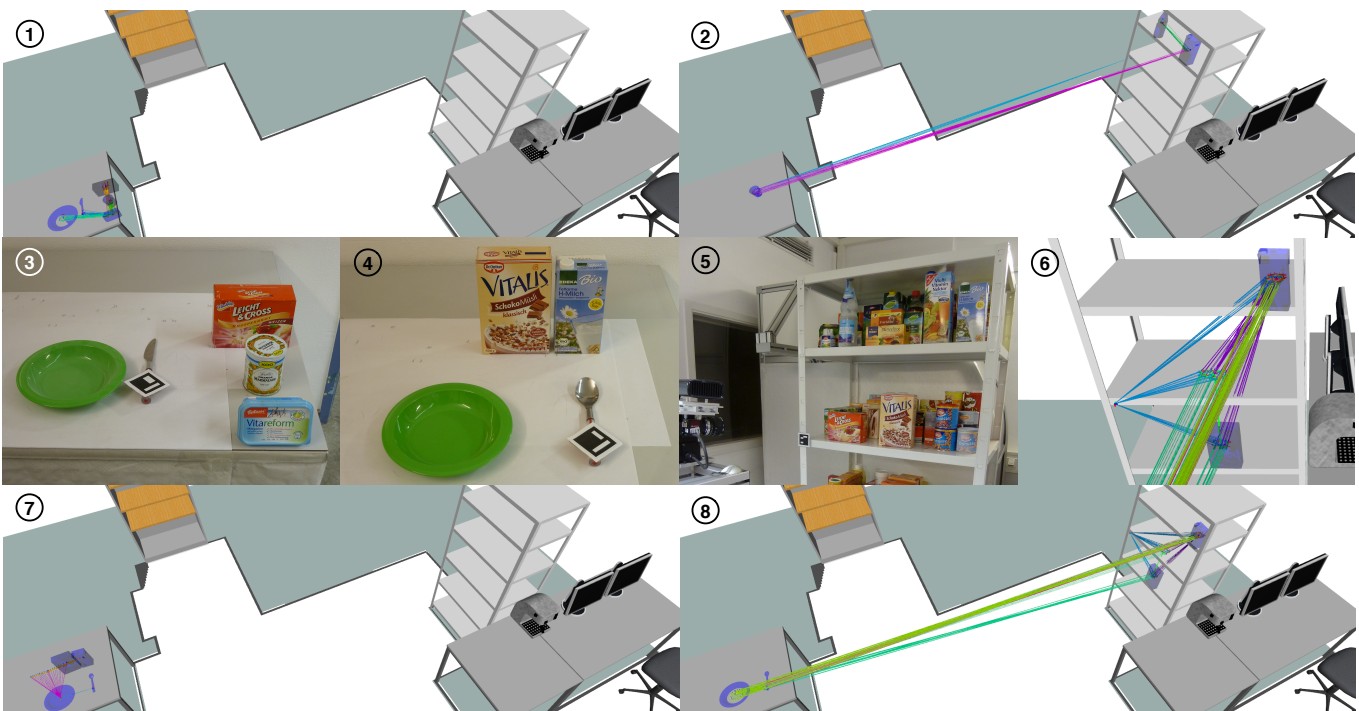

**Figure 14.** Object trajectories demonstrated for the scene categories "Sandwich-Setting" (images **1**, **3**), "Drinks-on Shelf" (images **2**, **5**), "Cereals-Setting" (images **4**, **7**), and "Cereals-on Shelf" (images **6**, **8**) in our kitchen setup, as well as the spatial relations inside the ISM trees learned for these categories.

5.2.5. Performance of Optimized ISM Trees

Table 3 shows how the ISM trees for the scene categories from the previous subsection performed concerning the goodness measures numFPs() (given as a percent) and avgDur(). One line corresponds to one category. The two measures were defined in Section 3.8 for the RTS. We rate their values with color coding, as in Section 5.2.2. Additionally, the table specifies the trajectory lengths and numbers of objects for each category, as well as the

number of relations it models. The presented values confirm that the runtime of the scene recognition with a complete topology was orders of magnitude higher than that with a star. Especially for the larger categories in Table 3, complete topologies were found to be much too inefficient for ASR. It should be noted that high runtimes do not only result from large numbers of objects, but also from long demonstration recordings. This, e.g., explains the runtime difference between "Setting-Ready for Breakfast" and "Cupboard-Filled". The table also displays the high numbers of false positives produced with star topologies, thus meaning that they are not an alternative to complete topologies. However, the number of false positives also depends on how much the objects in a category have been moved during a demonstration. If the objects are barely moved, such as was the case for "Sandwich-Setting", a star topology is just as reliable as a complete topology. Overall, however, only optimized topologies achieved simultaneously low values for numFPs() and for avgDur().

In Figure 15, we now consider the runtime of the scene recognition with ISM trees from optimized topologies individually. This plot shows the average runtimes for different datasets depending on the number of objects they contained and the length of the record of their demonstration. Unlike Table 3, which presents results from sensor-recorded demonstrations, all the object trajectories for this plot were generated via simulation. The scene recognition was performed on 600 object configurations generated for each dataset, which are shown in Figure 15. The recognition runtimes for these configurations are given in seconds, and each curve stands for a specific trajectory length. All curves appear to be linear for the number of objects. The slopes of the curves appear to be determined by the trajectory lengths. Thus, the runtimes appear to correlate with the product of the trajectory length and the number of objects. Beyond such a favorable time complexity, another experiment rendered ISM trees suitable for object search applications: We measured a maximum runtime of 3.71 s for ten objects and a trajectory length of 400 samples (20 min when capturing samples every 3 s while a demonstration was recorded). The fact that the runtimes with optimized topologies in Table 3 remained under 2.5 s further emphasizes that ISM trees are suitable for the state machine we used to implement ASR.

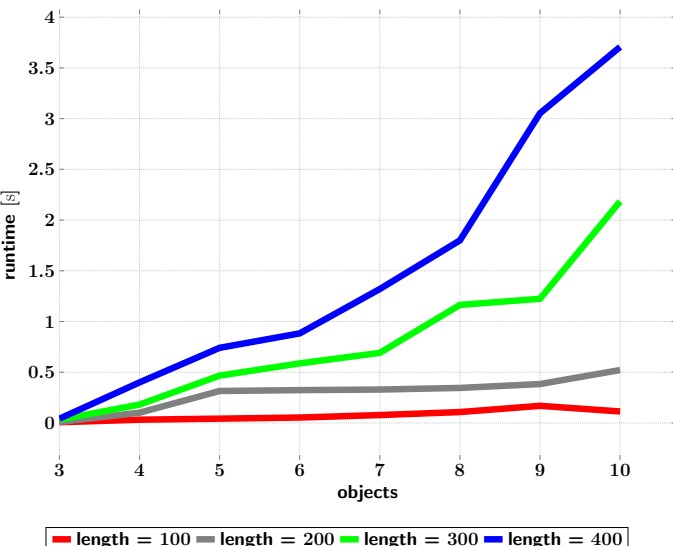

**Figure 15.** Times taken for ISM trees (from optimized topologies) to recognize scenes depending on the trajectory lengths in their datasets.

*5.3. Evaluation of Active Scene Recognition*

5.3.1. Influence of Object Orientations on Pose Prediction

Unlike Section 5.2.3, in this section, we no longer detect objects from a single viewpoint. Here, we investigated how well our ASR approach could recognize scenes whose object configurations cannot be fully perceived from a single viewpoint. We evaluated ASR in our

kitchen setup in six experiments. Each experiment was performed twice to account for the positioning uncertainties of our MILD robot shown in Figure 1. In these experiments, the robot was expected to recognize all the existing instances of the scene categories specified in Table 4. The experiments in Sections 5.3.1 and 5.3.2 analyze how our object pose prediction, and thus ASR, was affected when we changed the object poses between the previous demonstration and the execution of ASR during the experiment. Section 5.3.1 focuses on rotational displacements, whereas Section 5.3.2 addresses translational displacements.

**Table 4.** Performance measures for experiments with the physical MILD robot, which were used to evaluate active scene recognition. Color coding as in Table 1.

| Task | Duration [s] | CameraViews | Found Objects [%] | Confidences | | | | | | |
|------|-------------|-------------|-------------------|-------------|---------|----------|---------|----------|---------|----------|
| | | | | Setting | Drinks | | Cereals | | Sandwich | |
| | | | | Ready | Setting | on Shelf | Setting | on Shelf | Setting | on Shelf |
| s1_e1 | 783.96 | 16 | 100 | 0.97 | 0.33 | 0.99 | 0.49 | 0.85 | 0.99 | 0.34 |
| | 425.63 | 9 | 100 | 0.99 | 0.47 | 0.97 | 0.48 | 0.99 | 1.00 | 0.37 |
| s1_e2 | 560.18 | 12 | 100 | 0.97 | 0.33 | 0.42 | 0.46 | 0.82 | 0.99 | 0.49 |
| | 562.46 | 14 | 100 | 0.97 | 0.33 | 0.41 | 0.47 | 0.89 | 0.99 | 0.49 |
| s2_e1 | 367.32 | 8 | 100 | 0.93 | 0.33 | 0.98 | 0.96 | 0.75 | 1.00 | 0.45 |
| | 336.18 | 8 | 100 | 0.91 | 0.33 | 0.98 | 0.96 | 0.74 | 0.99 | 0.50 |
| s2_e2 | 584.07 | 13 | 100 | 0.95 | 0.33 | 0.73 | 0.97 | 0.68 | 0.99 | 0.34 |
| | 434.99 | 10 | 100 | 0.92 | 0.33 | 0.75 | 0.98 | 0.69 | 1.00 | 0.42 |
| s3_e1 | 533.29 | 11 | 93.75 | 0.99 | 0.93 | 0.25 | 0.99 | 0.50 | 1.00 | 0.34 |
| | 439.61 | 8 | 100 | 0.99 | 0.91 | 0.48 | 0.98 | 0.75 | 1.00 | 0.37 |

In both of the subsections, we investigated how well ASR could detect all objects in scene category instances despite displacements and whether it could detect the deviations from learned relations that result from these displacements. The first experiments (s1_e1 and s2_e1) in each subsection were performed on object configurations that were identical to the demonstration, whereas the second experiments (s1_e2 and s2_e2) covered either rotational or translational displacements. At the beginning of s1_e1 and s1_e2, MILD looked at the upper right corner of the shelves in **1** in Figure 16. Image **1** visualizes the results of one of the two executions of s1_e1 shown in Table 4. From there, MILD detected two searched objects. It then predicted object poses on the shelves and the table. NBV estimation minimizes the travel time for MILD by letting ASR search for another object on the shelves. Video S3 ("Influence of Object Orientations on Pose Prediction") shows how MILD proceeded further. In the end, the instances of the categories "Drinks-on Shelf", "Cereals-on Shelf", "Sandwich-Setting", and "Setting-Ready for Breakfast" were correctly recognized.

As is visible in **2**, all of the searched objects on the shelves were rotated for s1_e2. This change affected the categories "Drinks-on Shelf" and "Cereals-on Shelf". The fact that the confidence levels for the two categories fell off differently seems counterintuitive, but it is because they do not contain the same number of objects. The results of s1_e2 also shows that ASR requires only a single correctly oriented object (here, the shelves) to find all of the objects from the same scene category. This is because the shelves cause correctly predicted poses on the table, so ASR can ignore the more distant incorrect predictions at the lower left of **3** that result from the rotated objects. The views that the robot has just adopted and will adopt next are visualized by red and turquoise frustums, respectively. Predicted poses that are within the future view are colored blue. The large gap between correct and incorrect pose predictions illustrates the extent to which rotational changes can affect pose prediction when using long spatial relations.

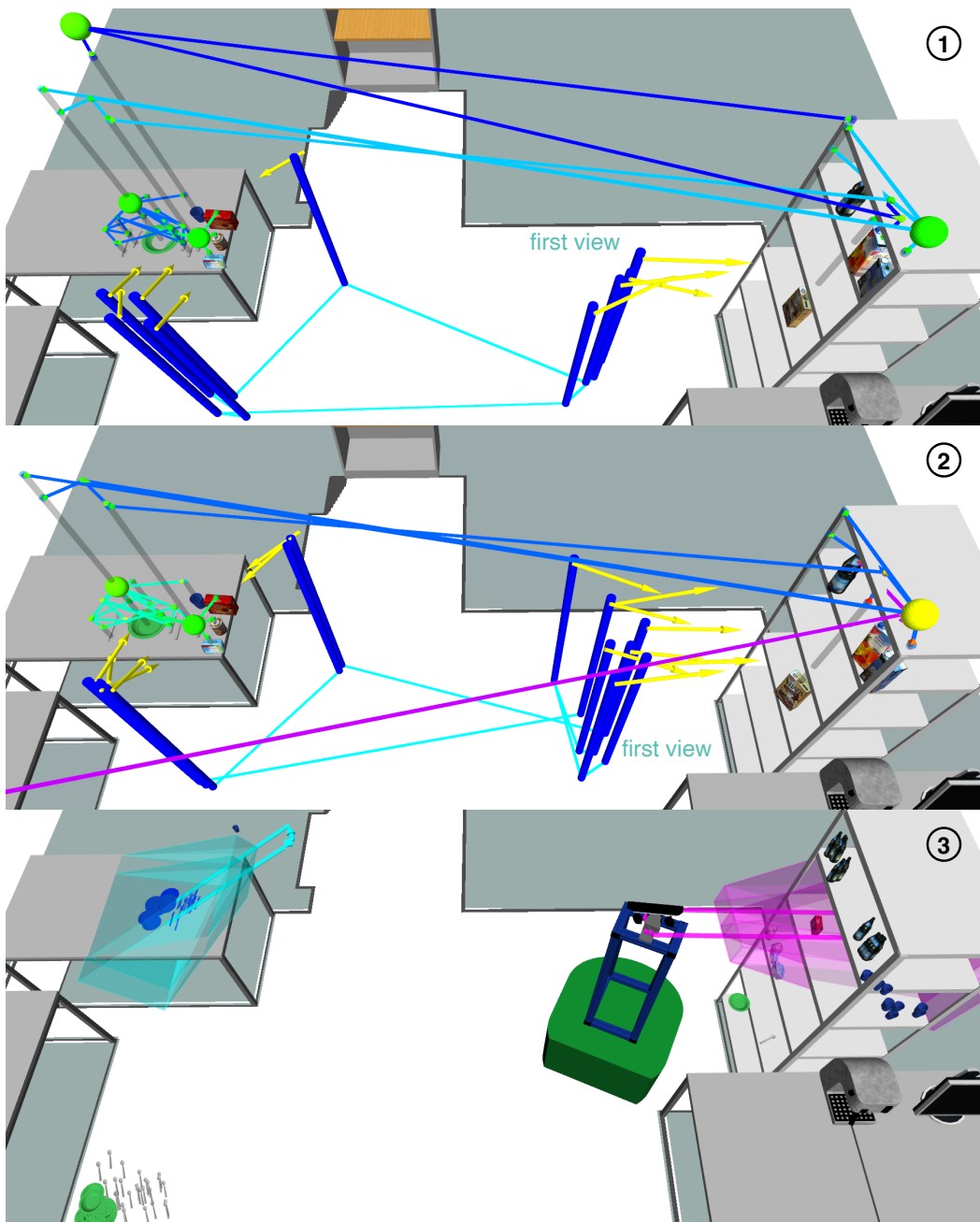

**Figure 16.** Influence of object orientations on pose prediction: Images **1** and **2** show s1_e1 and s1_e2, respectively. Between s1_e1 and s1_e2, all objects on the shelves (right) were rotated. Images **1** and **2**: Recognized scenes and camera views MILD adopted. Image **3**: Snapshot during the execution of s1_e2. The miniature objects correspond to predicted poses.

### 5.3.2. Influence of Object Positions on Pose Prediction

At the beginning of s2_e1, MILD was not standing in front of the shelves, but in front of the table. As is shown in **1** in Figure 17, MILD first searched this table. We have recorded in Video S4 ("Influence of Object Positions on Pose Prediction") how MILD proceeded until all existing scene categories were recognized. The difference between s2_e1 and s2_e2 is that all of the objects on the table were shifted at the same time using a tray. The confidence levels of those categories in Table 4, which contain only the objects on the table, remained unchanged. Since all of the objects were on the tray, shifting the tray did not affect the relative poses between them. This shows that ISM trees depend only on relative object poses and not directly on absolute object poses, which is thanks to the relations used.

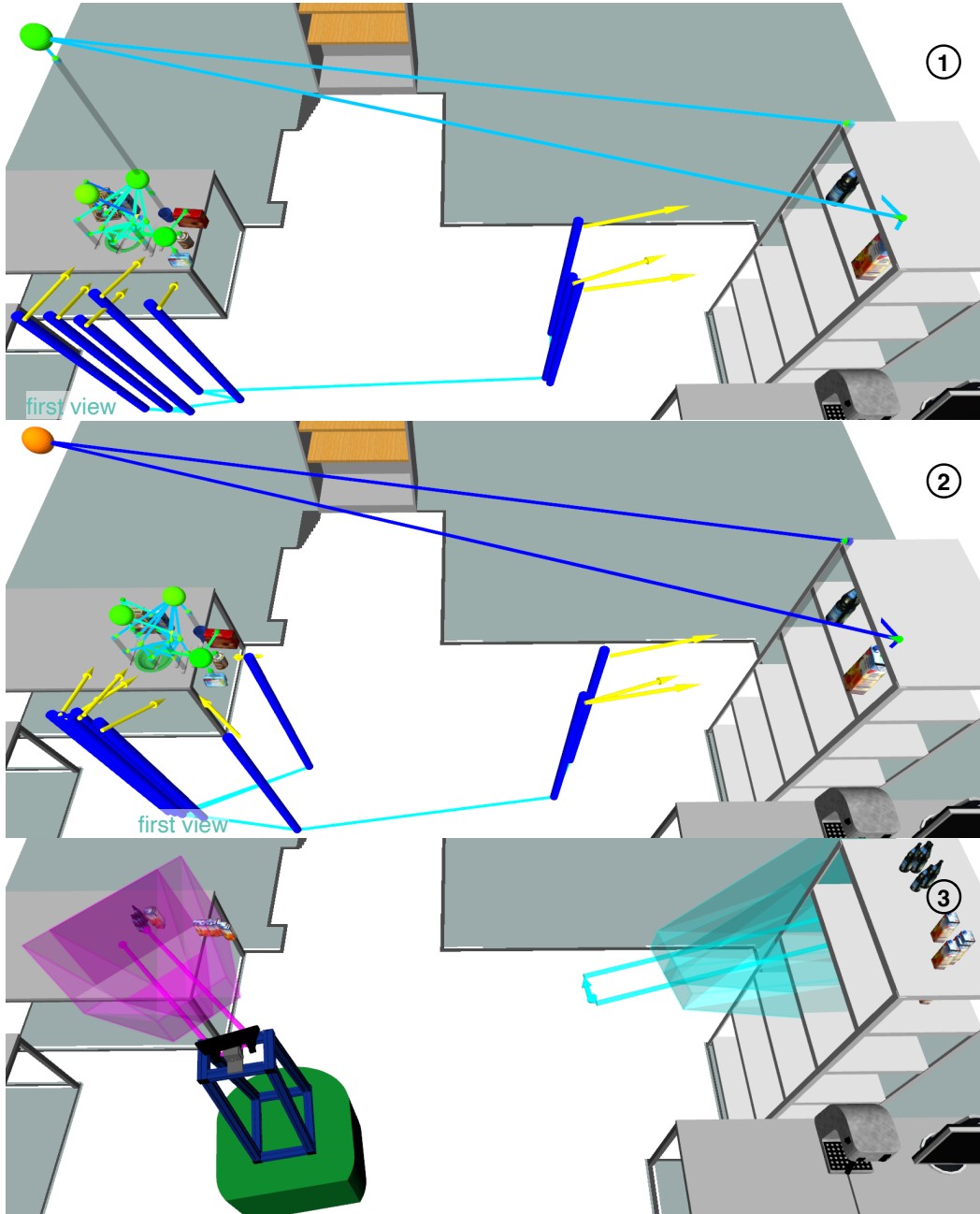

**Figure 17.** Influence of object positions on pose prediction: Images **1** and **2** show s2_e1 and s2_e2, respectively. Between s2_e1 and s2_e2, all objects on the table were shifted to the right. Images **1** and **2**: Recognized scenes and camera views MILD adopted. Image **3**: Snapshot during the execution of s2_e2. The miniature objects correspond to predicted poses.

The long relations between the objects on the table and those on the shelves were affected by the shift, but only slightly. The predicted poses on the shelves moved backwards, as are shown in **2** in Figure 17. However, they stayed close enough to the shelves, so MILD still found all of the objects. Yet, even a small orientation error in an object estimate on the table caused some predicted poses on one shelf to move up one level, as is shown on the right in **3**. Overall, the effect of rotational deviations on the accuracy of pose prediction depends on the length of the relation used, while that of translational deviations is constant.

### 5.3.3. Active Scene Recognition on a Cluttered Table

After two subsections devoted to object configurations spread across our kitchen setup, this subsection shows how our ASR approach delt with an object configurations that brought together a large number of searched objects from different overlapping scenes. Such a configuration—a cluttered table—can be seen in **1** in Figure 18. It consists of 15 objects to be searched, several of which are obscured from certain viewpoints, and 7 irrelevant objects. As **2** in Figure 18 and Video S5 ("Active Scene Recognition on a Cluttered Table") show, MILD managed to find all of the searched objects. The only object that was not always found in the corresponding experiment s3_e1 was the set of shelves, and this set did not participate in s3_e1. The objects at the front of the table were easily localized, and the scene recognition achieved high confidence levels, e.g., for "Setting-Ready for Breakfast". The objects in the back were more difficult to find, thus resulting in a lower confidence level for "Drinks-Setting". The spuriously high confidence level for "Cereals-on Shelf" resulted from a false positive returned by the object localization. All of the irrelevant objects were correctly discarded by ASR.

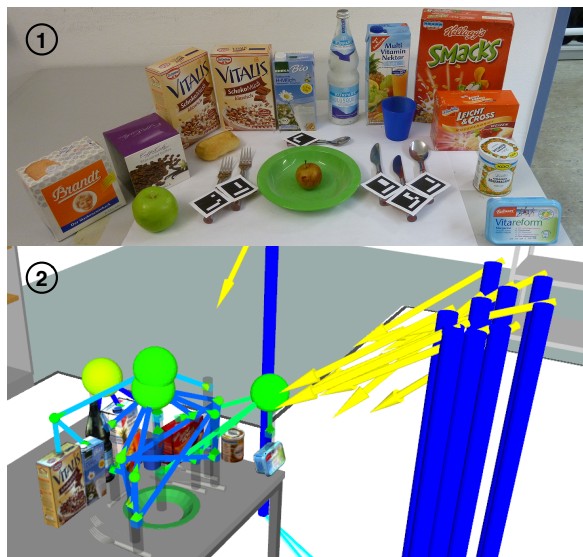

**Figure 18.** Active scene recognition on a cluttered table. Image **1**: Snapshot of physical objects. Image **2**: Camera views adopted and recognized scene instances.

### 5.3.4. Comparison of Three Approaches to ASR

In this subsection, we compare the execution times of our approach to ASR to those achieved by two alternative approaches to ASR. The searched object configuration looks similar to "Setting-Ready for Breakfast". The first alternative to our approach is called "direct search only" and omits the INDIRECT_SEARCH mode. Instead, its SCENE_RECOGNITION substate exclusively processes object estimates acquired by the informed and uninformed strategies of the DIRECT_SEARCH mode. We call the second alternative "bounding box search". This approach assumes that objects can only be located in so-called bounding boxes determined by prior knowledge and does not use INDIRECT_SEARCH to predict the poses of searched objects. Such bounding boxes are visualized in **3** in Figure 19 as yellow boxes in which possible object poses are visualized as white spheres. However, "bounding box search" uses next-best-views (NBVs) to sweep the bounding boxes.

Compared to the previous experiments (see **1** in Figure 18), the searched place setting has been shifted and rotated (see **2–4** in Figure 19). All three ASR approaches were executed twice and successfully found all of the objects in the setting. Since "direct search only" and "bounding box search" took an inordinate amount of time, we ran this experiment via simulation. Both alternatives took much longer than our ASR approach: 31.13 and 15.24 min instead of 2.48 min. The informed strategy of "direct search only" was not able to find all

objects. The views the strategy adopted are shown in **1** in Figure 19. MILD then used the uninformed strategy of "direct search only", which caused a lengthy search but eventually succeeded. Image **2** shows the views adopted by both the informed and uninformed strategies. The views in **3** were adopted by "bounding box search" and show that this approach often moves between bounding boxes rather than searching for a single one from different perspectives. This highlights the difficulty of parameterizing the estimation of NBVs. As seen in **4**, our approach adopted the same first two views as "direct search only" in **1**. However, instead of continuing the search on the right of the table once some objects were found, our approach adapted to the fact that the place setting had been rotated and let MILD search on the other side of the table.

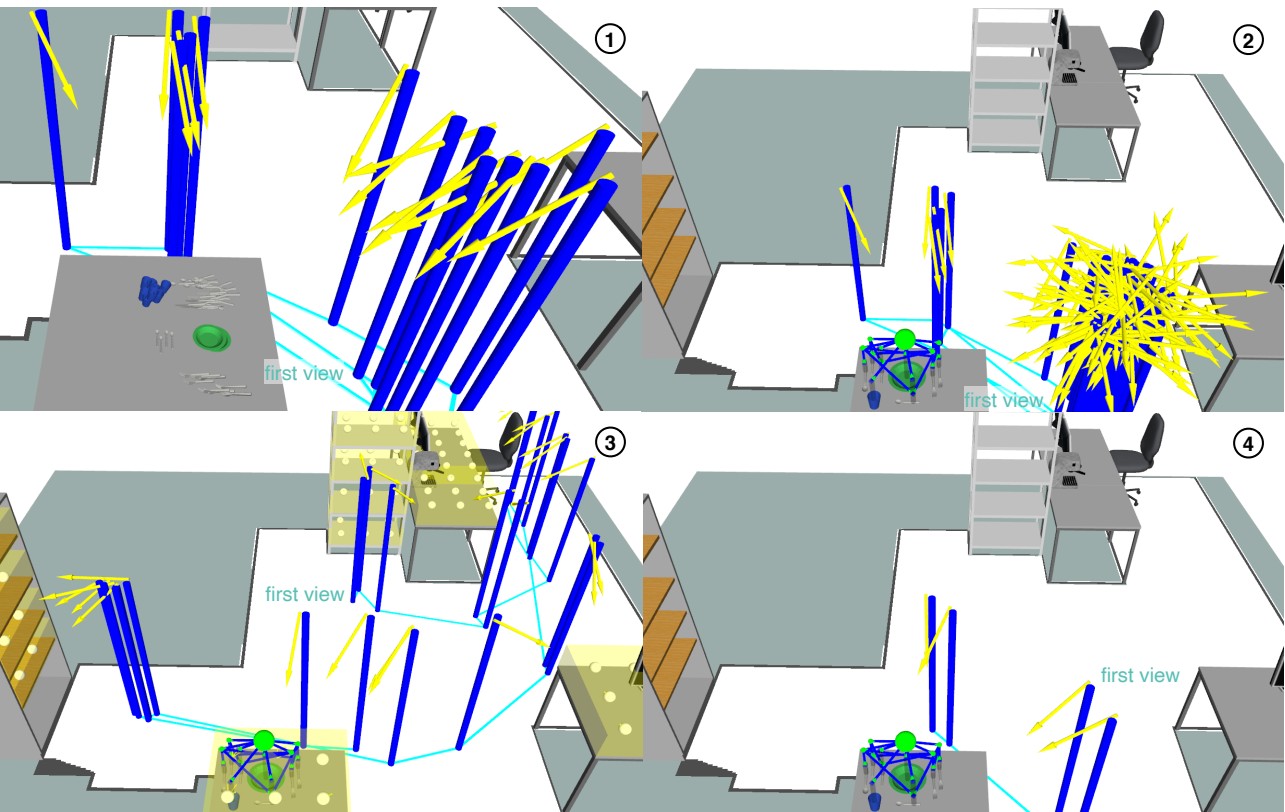

**Figure 19.** Comparison of three approaches to ASR: For "direct search only" (images **1** and **2**), "bounding box search" (image **3**), and our ASR approach (image **4**), we show which camera views were adopted. Additionally, image **1** shows demonstrated object poses, and images **2**–**4** show recognized scene instances.

### 5.3.5. Runtimes of Object Pose Prediction

We reused the ISM trees from Section 5.2.5 to compute the runtimes of our pose prediction algorithm for datasets with different numbers of objects $n$ and trajectory lengths $l$. We averaged $10 \cdot n \cdot l$ executions of the algorithm per dataset in Figure 20. Each runtime shown corresponds to the time required to predict the poses of all the objects in a category. If we disregard scaling (runtimes are given here in hundredths of a second), the analysis of the curves from Section 5.2.5 also applies to Figure 20. Given that the maximum runtime was 0.055 s for ten objects and a trajectory length of 400 samples, the time consumption of the object pose prediction seems negligible compared to the one of the scene recognition.

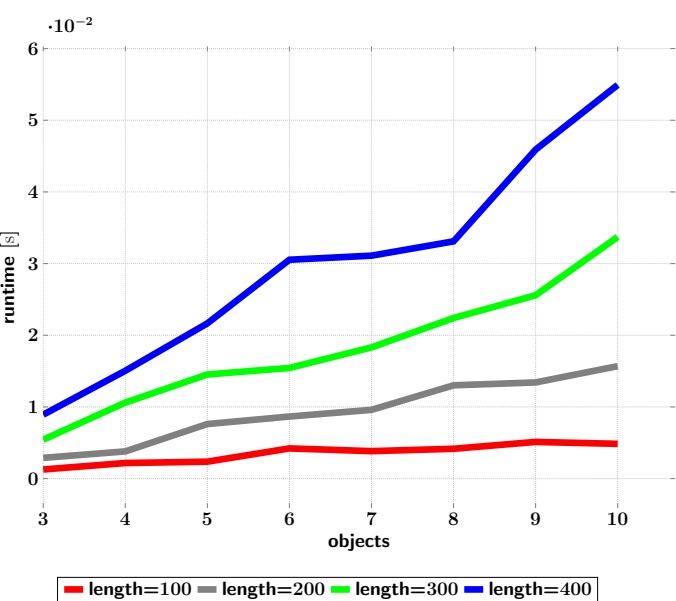

**Figure 20.** Times that our pose prediction algorithm took depending on the number of objects and trajectory length in the datasets.

## 6. Conclusions

Through its contributions, this article closes three gaps in the core of active scene recognition (ASR). ASR, which was impractical without these contributions, combines the scene recognition and object search—two tasks that are otherwise considered separately. Firstly, ASR enables the scene recognition to analyze object configurations that cannot be perceived from a single viewpoint. Secondly, it allows the object search to be guided by object configurations rather than single objects, thus making it more efficient. Using only single objects can lead to ambiguities, since, e.g., a knife in a table setting would expect a plate to be beneath itself when a meal is finished, while it would expect the plate to be beside itself when the meal has not yet started.

The feature extraction components of part-based models may be outdated compared to convolutional neural nets (today's gold standard). However, this article aims to show that ISMs are nevertheless particularly suitable for modeling the spatial characteristics of relations and their uncertainties. ISM trees additionally overcome the limitation of single ISMs to represent only a single relation topology. By replacing the feature extraction of ISMs with appropriate object pose estimators, ISM trees provide up-to-date object-based scene classification. Therefore, they can be seen as a complement to convolutional neural nets. Especially when modeling relations in scenes that express personal preferences and for which only small amounts of data are available, using techniques such as ISMs is suitable. This suitability stems from the fact that ISMs model relations nonparametrically in the sense of instance-based learning ([61]).

However, the fact that ISM trees model relations nonparametrically also means that they can be prone to combinatorial explosion. To avoid such effects in the recognition and prediction algorithms we contribute, we have implemented the following strategies: As proposed by [36], an accumulator array and a method similar to mean shift search were used within single ISMs during recognition to prune a significant portion of the votes. Moreover, when recognizing scenes with an ISM tree, two factors—the number of intermediate results passed from one ISM to the next and the lengths of the chains of interrelated ISMs in a tree—can cause combinatorial effects. We limited the first through passing only the best-rated intermediate results between ISMs and the second by minimizing the heights of the ISM trees through our tree generation algorithm. Moreover, we solved the combinatorial explosion that made our previous pose prediction algorithm inefficient. Instead of simply concatenating the inverted spatial relations in an ISM tree, the new method samples random subsets from these relations.

Our evaluation of PSR in Section 5.2 provided evidence that any ISM of an ISM tree precisely detects when object poses deviate from modeled spatial relations. Depending on their parametrization, ISMs are more or less permissive concerning such deviations. Further experiments in [9] have also shown that ISM trees are robust against objects that are missing in object configurations. In Section 5.3, we applied ASR to object configurations which were considerably more complex than those used in our previous work. Robot localization and object pose estimation accuracy were the limiting factors for our ASR approach. Still, ASR even succeeded in recognizing scenes at different locations with the same ISM tree. This illustrates that spatial relations make ISM trees particularly reusable compared to techniques that model scenes using absolute object poses. Video S6 ("Recognition of scenes independent of their locations") is devoted to this major advantage of ISM trees and thus ASR. The experiments in Section 5.2.5 suggest that the runtimes of PSR scenarios linearly depend on the number of objects included in a scene category. An experiment we conducted for datasets including six objects indicates that this is also true for trajectory lengths: We measured a maximum recognition runtime of 4.94 s for a demonstration trajectory length of 1000 samples. Still, the recognition runtimes for longer trajectory lengths can exceed the requirements of ASR. To overcome this limitation, we plan to compress the relations in ISM trees by eliminating redundant relative poses using downsampling voxel grids.

**Supplementary Materials:** The following supporting information can be downloaded at: https://www.mdpi.com/article/10.3390/robotics12060158/s1, Video S1: Demonstration of object configuration for learning a scene classifier; Video S2: Demonstration of the scene category—Office; Video S3: Influence of object orientations on pose prediction; Video S4: Influence of object positions on pose prediction; Video S5: Active scene recognition on a cluttered table; Video S6: Recognition of scenes independent of their locations.

**Author Contributions:** Conceptualization, P.M. and R.D.; methodology, P.M.; software, P.M.; validation, P.M.; formal analysis, P.M.; investigation, P.M.; resources, P.M: and R.D.; data curation, P.M.; writing—original draft preparation, P.M.; writing—review and editing, P.M. and R.D.; visualization, P.M.; supervision, R.D.; project administration, P.M. and R.D.; funding acquisition, P.M. and R.D. All authors have read and agreed to the published version of the manuscript.

**Funding:** This research was funded by the Deutsche Forschungsgemeinschaft via grant number 255319423. The APC was funded by the Technical University of Applied Sciences Wuerzburg-Schweinfurt (THWS).

**Data Availability Statement:** The source code for the presented active scene recognition system is publicly available at http://wiki.ros.org/asr (accessed on 1 June 2023).

**Acknowledgments:** This work draws from student projects by Fabian Hanselmann, Heinreich Heizmann, Oliver Karrenbauer, Felix Marek, Jonas Mehlhaus, Patrick Stöckle, Reno Reckling, Daniel Stroh, and Jeremias Trautmann. Our special thanks go to Rainer Jäkel, Michael Beetz, and Torsten Kröger for their valuable advice.

**Conflicts of Interest:** The authors declare no conflict of interest. The funding sponsors had no role in the design of the study; in the collection, analyses, or interpretation of data; in the writing of the manuscript, or in the decision to publish the results.

## Appendix A

This subsection provides pseudocodes for the three contributions of this article. Algorithm A1 is our algorithm for generating ISM trees and corresponds to contribution 1. Algorithms A2–A4 form our algorithm for scene recognition using ISM trees and correspond to contribution 2. Algorithms A5 and A6 form our algorithm for predicting object poses with ISM trees and correspond to contribution 3. As it is not the goal of this article to describe all the details of these algorithms, but to present their key ideas concisely, some variables and helper functions are only defined in [9].

---

**Algorithm A1** generateISMTreeFromStarTopologies$(z, \{\Sigma_\sigma\}, \{\mathbf{J}(o)\}) \to \{m\}$.

---

1: $i \leftarrow |\{\Sigma_\sigma\}| - 2$ and parentFound $\leftarrow$ **false**
2: $h_{\{\Sigma_\sigma\}} \leftarrow \max\limits_{o \in \{o\}} h_{\{\Sigma_\sigma\}}(o)$
3: **for** $h_j \leftarrow h_{\{\Sigma_\sigma\}}, \ldots, 0$ **do**
4:     **for all** $\Sigma_\sigma(j) \in \{\Sigma_\sigma(j) | \Sigma_\sigma(j) \in \{\Sigma_\sigma\} \wedge \exists o \in \ H_D(\Sigma_\sigma(j)) : h_{\{\Sigma_\sigma\}}(o) = h_j\}$ **do**
5:         Randomly extract $o_H$ from $\underset{o \in H_D(\Sigma_\sigma(j))}{\arg\min} \ h_{\{\Sigma_\sigma\}}(o)$
6:         **if** $h_j = 0$ **then**
7:             $z_m \leftarrow z$
8:         **else**
9:             $z_m \leftarrow$ append(append($z, ''\_sub''), i$)
10:             $i \leftarrow i - 1$
11:         $(m, \mathbf{J}_F) \leftarrow$ learnISM$(z_m, \mathbf{J}(o_M(\Sigma_\sigma(j))) \cup \{\mathbf{J}(o)|$
           $\mathbf{J}(o) \in \{\mathbf{J}(o)\} \wedge o \in N(o_M(\Sigma_\sigma(j)))\})$
12:         Create $o_F$ with $\mathbf{J}_F$ as its trajectory
13:         $\{m\} \leftarrow \{m\} \cup m$ and $\{\Sigma_\sigma\} \leftarrow \{\Sigma_\sigma\} \setminus \Sigma_\sigma(j)$
14:         **for** $h_k \leftarrow 0, \ldots, h_j - 1$ **do**
15:             **for all** $\Sigma_\sigma(k) \in \{\Sigma_\sigma(k) | \Sigma_\sigma(k) \in \{\Sigma_\sigma\} \wedge \exists o \in \ H_D(\Sigma_\sigma(k)) : h_{\{\Sigma_\sigma\}}(o) = h_k\}$ **do**
16:                 **if** $o_H \in N(o_M(\Sigma_\sigma(k)))$ **then**
17:                     $N(o_M(\Sigma_\sigma(k))) \leftarrow (N(o_M(\Sigma_\sigma(k))) \setminus o_H) \cup o_F$
18:                     $\{\mathbf{J}(o)\} \leftarrow \{\mathbf{J}(o)\} \cup \mathbf{J}_F$ and $h_{\{m\}}(m) \leftarrow h_k$
19:                     parentFound $\leftarrow$ **true**
20:                     **break**
21:             **if** parentFound = **true then**
22:                 parentFound $\leftarrow$ **false**
23:                 **break**
24: **return** $\{m\}$

---

**Algorithm A2** evaluateISMsInTree$(\{i\}, \{m\}) \to \{\mathbf{I}_{\{m\}}\}$.

---

1: **for** $h \leftarrow h_{\{m\}}, \ldots, 1$ **do**
2:     **for all** $\{m | m \in \{m\} \wedge h_{\{m\}}(m) = h\}$ **do**
3:         $\{\mathbf{I}_m\} \leftarrow$ recognitionSingleISM$(\{i\}, m)$
4:         **for all** $\mathbf{I}_m \in \{\mathbf{I}_m\}$ **do**
5:             Create $o_F$ with $\mathbf{E}(o_F) = (z, 0, \mathbf{T}_F)$ and $b(o_F) = b_F$, all extracted from $\mathbf{I}_m$
6:             $\{i\} = \{i\} \cup \{o_F\}$
7:             $\{\mathbf{I}_{\{m\}}\} = \{\mathbf{I}_{\{m\}}\} \cup \{\mathbf{I}_m\}$
8: $\{\mathbf{I}_{m_R}\} \leftarrow$ recognitionSingleISM$(\{i\}, m)$
9: $\{\mathbf{I}_{\{m\}}\} = \{\mathbf{I}_{\{m\}}\} \cup \{\mathbf{I}_{m_R}\}$
10: **return** $\{\mathbf{I}_{\{m\}}\}$

---

**Algorithm A3** assembleInstances$(\{\mathbf{I}_{\{m\}}\}, \epsilon_R) \to \{\mathbf{I_S}\}$.

---

1: **for all** $\mathbf{I}_{m_R} \in \{\mathbf{I}_{\{m\}}\}$ **do**
2:     **if** $b(\mathbf{I}_{m_R}) \geq \epsilon_R$ **then**
3:         $\{\mathbf{I}\} \leftarrow$ findSubInstances$(\mathbf{I}_{m_R}, \{\mathbf{I}_{\{m\}}\})$
4:         $\mathbf{I_S} \leftarrow \{\mathbf{I}\} \cup \mathbf{I}_{m_R}$
5:         $\{\mathbf{I_S}\} \leftarrow \{\mathbf{I_S}\} \cup \mathbf{I_S}$
6: **return** $\{\mathbf{I_S}\}$

---

**Algorithm A4** findSubInstances($\mathbf{I}_{m'}, \{\mathbf{I}_{\{m\}}\}) \rightarrow \{\mathbf{I}\}$.

---

1:   Extract $\{i\}_{m'}$ from $\mathbf{I}_{m'}$
2:   **for all** $i \in \{i\}_{m'}$ **do**
3:      Extract $c$ and $\mathbf{T}$ from $\mathbf{E}(i)$
4:      **for all** $\mathbf{I}_{\{m\}} \in \{\mathbf{I}_{\{m\}}\}$ **do**
5:         Extract $z$ and $\mathbf{T}_F$ from $\mathbf{I}_{\{m\}}$
6:         **if** $c = z \wedge \mathbf{T} = \mathbf{T}_F$ **then**
7:            $\{\mathbf{I}_t\} \leftarrow \{\mathbf{I}_t\} \cup \mathbf{I}_{\{m\}}$
8:   **for all** $\mathbf{I}_t \in \{\mathbf{I}_t\}$ **do**
9:      $\{\mathbf{I}\} \leftarrow \{\mathbf{I}\} \cup$ findSubInstances($\mathbf{I}_t, \{\mathbf{I}_{\{m\}}\}$)
10:     $\{\mathbf{I}\} \leftarrow \{\mathbf{I}\} \cup \mathbf{I}_t$
11:   **return** $\{\mathbf{I}\}$

---

**Algorithm A5** predictPose($o_P, \mathbf{P}^*_{\{m\}}(o_P), \mathbf{T}_F) \rightarrow \mathbf{T}_P$.

---

1:   $\mathbf{T}_P \leftarrow \mathbf{T}_F$
2:   **for** $k \leftarrow 1, \ldots, \left|\mathbf{P}^*_{\{m\}}(o_P)\right| + 1$ **do**
3:      **if** $k = \left|\mathbf{P}^*_{\{m\}}(o_P)\right| + 1$ **then**
4:         $o \leftarrow o_P$
5:      **else**
6:         $o \leftarrow o_F(m_{k+1})$ with $m_{k+1}$ in $\mathbf{P}^*_{\{m\}}(o_P)$
7:      $\mathbf{T}_P \leftarrow$ randomVoteOnPose($\mathbf{T}_P, m_k, o$)
8:   **return** $\mathbf{T}_P$

---

**Algorithm A6** generateCloudOfPosePredictions($\mathbf{I_S}, \mathbf{P}^*_{\{m\}}(o), n_P) \rightarrow \{\mathbf{T}_P(o)\}$.

---

1:   Extract $\mathbf{T}_F$ and $\{i\}_\mathbf{S}$ from $\mathbf{I_S}$
2:   **for all** $o_P \in \{o\} \setminus \{i\}_\mathbf{S}$ **do**
3:      **for** $i \leftarrow 0, \ldots, n_P$ **do**
4:         $\mathbf{T}_P \leftarrow$ predictPose($o_P, \mathbf{P}^*_{\{m\}}(o_P), \mathbf{T}_F$)
5:         $\{\mathbf{T}_P(o_P)\} \leftarrow \{\mathbf{T}_P(o_P)\} \cup \mathbf{T}_P$
6:      $\{\mathbf{T}_P(o)\} \leftarrow \{\mathbf{T}_P(o)\} \cup \{\mathbf{T}_P(o_P)\}$
7:   **return** $\{\mathbf{T}_P(o)\}$

---

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
