# Peer review of "Implicit Shape Model Trees: Recognition of 3-D Indoor Scenes and Prediction of Object Poses for Mobile Robots"

_robotics, doi:10.3390/robotics12060158_

Round 1
Reviewer 1 Report
Comments and Suggestions for Authors
This paper discusses three main contributions to the area of active scene recognition - algorithms for ISM generation, and scene recognition and object pose prediction using ISMs. It also discusses the benefits of leveraging part-based models, while swapping out their traditional feature extraction techniques for CNN-based feature extraction.
Overall this paper does a great job setting up the background, describing the problem and previous work, outlining the novel approaches, and providing experimental results and analysis. I think this paper is ready for publication as-is.
Author Response
Dear Reviewer
Thank you very much for your kind and encouraging feedback. It is most appreciated.
Kind regards
Pascal Meissner
Reviewer 2 Report
Comments and Suggestions for Authors
In general, the article is well written. The paper should interest enough for people who are interested in the subject of “mobile robotics and object scene recognition”. For this reason and based on the obtained results, I accept the paper for publication.

Comments on the Quality of English LanguageMinor editing of English language required
Author Response
Dear Reviewer
Thank you very much for the accurate summary and your feedback. We hope our revision addresses your concerns and requests.
Please find detailed responses to your comments in the attached PDF file.
Kind regards
Pascal Meissner

Reviewer 3 Report
Comments and Suggestions for Authors
The paper presents several improvements to previous work already published by the authors. These new contributions, although quite related to the previous work, are very relevant for improving the practical use of the method. The authors clearly identify throughout the paper which part is their previous work and which one corresponds to a new contribution.
I think that it will that the paper would be easier to follow if some figures and tables are grouped on the same page. As an example, I will cite Tables 1 and 2 and Figure 13 or Figures 6 and 7.
Author Response
Dear Reviewer
Thanks a lot for this recommendation, as well as for the time and effort invested in reviewing our work.
As you suggested, Figures 6 and 7 have been placed on the same page.
Tables 1 and 2 and Figure 13 have also been brought together on one page. The caption of Table 2 has been adjusted so that everything fits on the same page.
We hope that these changes address your concern.
Kind regards
Pascal Meissner